# Leveraging Labeled and Unlabeled Data for Consistent Fair Binary Classification

**Evgenii Chzhen[1,2], Christophe Denis[1], Mohamed Hebiri[1],**
**Luca Oneto[3], Massimiliano Pontil[4,5]**
[1]Université Paris-Est, [2]Université Paris-Sud, [3]University of Pisa,
[4]Istituto Italiano di Tecnologia, [5]University College London
evgenii.chzhen@math.u-psud.fr, {mohamed.hebiri,christophe.denis}@u-pem.fr,
luca.oneto@unipi.it, massimiliano.pontil@iit.it

## Abstract

We study the problem of fair binary classification using the notion of Equal Opportunity. It requires the true positive rate to distribute equally across the sensitive groups. Within this setting we show that the fair optimal classifier is obtained by recalibrating the Bayes classifier by a group-dependent threshold. We provide a constructive expression for the threshold. This result motivates us to devise a plug-in classification procedure based on both unlabeled and labeled datasets. While the latter is used to learn the output conditional probability, the former is used for calibration. The overall procedure can be computed in polynomial time and it is shown to be statistically consistent both in terms of the classification error and fairness measure. Finally, we present numerical experiments which indicate that our method is often superior or competitive with the state-of-the-art methods on benchmark datasets.

## 1 Introduction

As machine learning becomes more and more spread in our society, the potential risk of using algorithms that behave unfairly is rising. As a result there is growing interest to design learning methods that meet "fairness" requirements, see [5, 9, 10, 17, 19, 22–24, 28, 31, 33, 47, 48, 50, 52] and references therein. A central goal is to make sure that sensitive information does not "unfairly" influence the outcomes of learning methods. For instance, if we wish to predict whether a university student applicant should be offered a scholarship based on curriculum, we would like our model to not unfairly use additional sensitive information such as gender or race.

Several measures of fairness of a classifier have been studied in the literature [49], ranging from Demographic Parity [8], Equal Odds and Equal Opportunity [22], Disparate Treatment, Impact, and Mistreatment [48], among others. In this paper, we study the problem of learning a binary classifier which satisfies the Equal Opportunity fairness constraint. It requires that the true positive rate of the classifier is the same across the sensitive groups. This notion has been used extensively in the literature either as a postprocessing step [22] on a learned classifier or directly during training, see for example [17] and references therein.

We address the important problem of devising statistically consistent and computationally efficient learning procedures that meet the fairness constraint. Specifically, we make four contributions. First, we derive in Proposition 2.3 the expression for the optimal equal opportunity classifier, derived via thresholding of the Bayes regressor. Second, inspired by the above result we proposed a semi-supervised plug-in type method, which first estimates the regression function on labeled data and then estimates the unknown threshold using unlabeled data. Consequently, we establish in Theorem 4.5 that the proposed procedure is consistent, that is, it asymptotically satisfies the equal opportunity

constraint and its risk converges to the risk of the optimal equal opportunity classifier. Finally, we present numerical experiments which indicate that our method is often superior or competitive with the state-of-the-art on benchmark datasets.

We highlight that the proposed learning algorithm can be applied on top of any off-the shelf method which consistently estimates the regression function (class condition probability), under mild additional assumptions which we discuss in the paper. Furthermore, our calibration procedure is based on solving a simple univariate problem. Hence the generality, statistical consistency and computational efficiency are strengths of our approach.

The paper is organized in the following manner. In Section 2, we introduce the problem and derive a form of the optimal equal opportunity classifier. Section 3 is devoted to the description of our method. In Section 4 we introduce assumptions used throughout this work and establish that the proposed learning algorithm is consistent. Finally, Section 5 presents numerical experiments with our method.

## 1.1 Related work

In this section we review previous contributions on the subject. Works on algorithmic fairness can be divided in three families. Our algorithm falls within the first family, which modifies a pretrained classifier in order to increase its fairness properties while maintaining as much as possible the classification performance, see [6, 20, 22, 38] and references therein. Importantly, for our approach the post-processing step requires only unlabeled data, which is often easier to collect than its labeled counterpart. Methods in the second family enforce fairness directly during the training step, e.g. [2, 12, 17, 37]. The third family of methods implements fairness by modifying the data representation and then employs standard machine learning methods, see e.g. [1, 9, 17, 25–27, 50] as representative examples.

To the best of our knowledge the formula for the optimal fair classifier presented here is novel. In [22] the authors note that the optimal equalized odds or equal opportunity classifier can be derived from the Bayes optimal regressor, however, no explicit expression for this threshold is provided. The idea of recalibrating the Bayes classifier is also discussed in a number of papers, see for example [35, 38] and references therein. More importantly, the problem of deriving *efficient* and *consistent* estimators under fairness constraints has received limited attention in the literature. In [17], the authors present consistency results under restrictive assumptions on the model class. Furthermore, they only consider convex approximations of the risk and fairness constraint and it is not clear how to relate their results to the original problem with the miss-classification risk. In [2], the authors reduce the problem of fair classification to a sequence of cost-sensitive problems by leveraging the saddle point formulation. They show that their algorithm is consistent in both risk and fairness constraints. However, similarly to [17], the authors of [2] assume that the family of possible classifiers admits a bounded Rademacher complexity.

Plug-in methods in classification problems are well established and are well studied from statistical perspective, see [4, 16, 46] and references therein; in particular, it is known that one can build a plug-in type classifier which is optimal in minimax sense [4, 46]. Until very recently, theoretical studies on such methods were reduced to an efficient estimation of the regression function. Indeed, in standard settings of classification the threshold is always known beforehand, thus, all the information about the optimal classifier is wrapped into the distribution of the label conditionally on the feature.

More recently, classification problems with a distribution dependent threshold have emerged. Prominent examples include classification with non-decomposable measures [30, 45, 51], classification with reject option [15, 32], and confidence set setup of multi-class classification [11, 14, 40], among others. A typical estimation algorithm in these scenarios is based on the plug-in strategy, which uses extra data to estimate the unknown threshold. Interestingly, in some setups a practitioner does not need to have access to two labeled samples and optimal estimation can be efficiently performed in semi-supervised manner [11, 14].

## 2 Optimal Equal Opportunity classifier

Let $(X, S, Y)$ be a tuple on $\mathbb{R}^d \times \{0, 1\} \times \{0, 1\}$ having a joint distribution $\mathbb{P}$. Here the vector $X \in \mathbb{R}^d$ is seen as the vector of features, $S \in \{0, 1\}$ a binary sensitive variable and $Y \in \{0, 1\}$ a binary output label that we wish to predict from the pair $(X, S)$. We also assume that the distribution

is non-degenerate in $Y$ and $S$ that is $\mathbb{P}(S = 1) \in (0,1)$ and $\mathbb{P}(Y = 1) \in (0,1)$. A classifier $g$ is a measurable function from $\mathbb{R}^d \times \{0,1\}$ to $\{0,1\}$, and the set of all such functions is denoted by $\mathcal{G}$. In words, each classifier receives a pair $(x,s) \in \mathbb{R}^d \times \{0,1\}$ and outputs a binary prediction $g(x,s) \in \{0,1\}$. For any classifier $g$ we introduce its associated miss-classification risk as

$$\mathcal{R}(g) := \mathbb{P}\left(g(X,S) \neq Y\right) \ . \tag{1}$$

A *fair* optimal classifier is formally defined as

$$g^* \in \arg\min_{g \in \mathcal{G}} \left\{\mathcal{R}(g) \ : \ g \text{ is fair}\right\} \ .$$

There are various definitions of fairness available in the literature, each having its critics and its supporter. In this work, we employ the following definition introduced in [22]. We refer the reader to this work as well as [2, 17, 35] for a discussion, motivation of this definition, and a comparison to other fairness definitions.

**Definition 2.1** (Equal Opportunity [22])**.** *A classifier* $(x,s) \mapsto g(x,s) \in \{0,1\}$ *is called fair if*

$$\mathbb{P}\left(g(X,S) = 1 \,|\, S = 1, Y = 1\right) = \mathbb{P}\left(g(X,S) = 1 \,|\, S = 0, Y = 1\right) \ .$$

*The set of all fair classifiers is denoted by* $\mathcal{F}(\mathbb{P})$.

Note, that the definition of fairness depends on the underlying distribution $\mathbb{P}$ and hence the whole class $\mathcal{F}(\mathbb{P})$ of the fair classifiers should be estimated. Further, notice that the class $\mathcal{F}(\mathbb{P})$ is non-empty as it always contains a classifier $g(x,s) \equiv 0$.

Using this notion of fairness we define an optimal equal opportunity classifier as a solution of the optimization problem

$$\min_{g \in \mathcal{G}} \left\{\mathcal{R}(g) \ : \ \mathbb{P}\left(g(X,S) = 1 \,|\, Y = 1, S = 1\right) = \mathbb{P}\left(g(X,S) = 1 \,|\, Y = 1, S = 0\right)\right\} \ . \tag{2}$$

We now introduce an assumption on the regression function that plays an important role in establishing the form of the optimal fair classifier.

**Assumption 2.2.** *For each* $s \in \{0,1\}$ *we require the mapping* $t \mapsto \mathbb{P}\left(\eta(X,S) \leq t \,|\, S = s\right)$ *to be continuous on* $(0,1)$*, where for all* $(x,s) \in \mathbb{R}^d \times \{0,1\}$*, we let the regression function*

$$\eta(x,s) := \mathbb{P}\left(Y = 1 \,|\, X = x, S = s\right) = \mathbb{E}\left[Y \,|\, X = x, S = s\right] \ .$$

*Moreover, for every* $s \in \{0,1\}$*, we assume that* $\mathbb{P}\left(\eta(X,s) \geq 1/2 \,|\, S = s\right) > 0$.

The first part of Assumption 2.2 is achieved by many distributions and has been introduced in various contexts, see e.g. [11, 15, 32, 40, 45] and references therein. It says that, for every $s \in \{0,1\}$ the random variable $\eta(X,s)$ does not have atoms, that is, the event $\{\eta(X,s) = t\}$ has probability zero. The second part of the assumption states that the regression function $\eta(X,s)$ must surpass the level $1/2$ on a set of non-zero measure. Informally, returning to scholarship example mentioned in the introduction, this assumption means that there are individuals from *both* groups who are more likely to be offered a scholarship based on their curriculum.

In the following result we establish that the optimal equal opportunity classifier is obtained by recalibrating the Bayes classifier.

**Proposition 2.3** (Optimal Rule)**.** *Under Assumption 2.2 an optimal classifier* $g^*$ *can be obtained for all* $(x,s) \in \mathbb{R}^d \times \{0,1\}$ *as*

$$g^*(x,1) = \mathbf{1}_{\left\{1 \leq \eta(x,1)\left(2 - \frac{\theta^*}{\mathbb{P}(Y=1,S=1)}\right)\right\}}, \quad g^*(x,0) = \mathbf{1}_{\left\{1 \leq \eta(x,0)\left(2 + \frac{\theta^*}{\mathbb{P}(Y=1,S=0)}\right)\right\}} \tag{3}$$

*where* $\theta^* \in \mathbb{R}$ *is determined from the equation*

$$\frac{\mathbb{E}_{X|S=1}\left[\eta(X,1)\mathbf{1}_{\left\{1 \leq \eta(X,1)\left(2 - \frac{\theta^*}{\mathbb{P}(Y=1,S=1)}\right)\right\}}\right]}{\mathbb{P}(Y=1\,|\,S=1)} = \frac{\mathbb{E}_{X|S=0}\left[\eta(X,0)\mathbf{1}_{\left\{1 \leq \eta(X,0)\left(2 + \frac{\theta^*}{\mathbb{P}(Y=1,S=0)}\right)\right\}}\right]}{\mathbb{P}(Y=1\,|\,S=0)} \ .$$

*Furthermore it holds that* $|\theta^*| \leq 2$.

*Proof sketch.* The proof relies on weak duality. The first step of the proof is to write the minimization problem for $g^*$ using a "min-max" problem formulation. We consider the corresponding dual "max-min" problem and show that it can be analytically solved. Then, the continuity part of Assumption 2.2 allows to demonstrate that the solution of the "max-min" problem gives a solution of the "min-max" problem. The second part of Assumption 2.2 is used to prove that $|\theta^*| \leq 2$. $\qquad\square$

Before proceeding further, let us define a notion of unfairness, which plays a key role in our statistical analysis; it is sometimes referred to as difference of equal opportunity (DEO) in the literature [see e.g. 17].

**Definition 2.4** (Unfairness). *For any classifier g we define its unfairness as*

$$\Delta(g, \mathbb{P}) = |\mathbb{P}\left(g(X, S) = 1 \,|\, S = 1, Y = 1\right) - \mathbb{P}\left(g(X, S) = 1 \,|\, S = 0, Y = 1\right)| \ .$$

A principal goal of this paper is to construct a classification algorithm $\hat{g}$ which satisfies

$$\underbrace{\mathbb{E}[\Delta(\hat{g}, \mathbb{P})] \to 0,}_{\text{asymptotically fair}} \quad \text{and} \quad \underbrace{\mathbb{E}[\mathcal{R}(\hat{g})] \to \mathcal{R}(g^*)}_{\text{asymptotically optimal}} \ ,$$

where the expectations are taken with respect to the distribution of data samples. As we shall see our estimator is built from independent sets of labeled and unlabeled samples. Hence the convergence above is meant to hold as both samples grow to infinity.

## 3 Proposed procedure

In this section, we present the proposed plug-in algorithm and begin to study its theoretical properties.

We assume that we have at our disposal two datasets, labeled $\mathcal{D}_n$ and unlabeled $\mathcal{D}_N$ defined as

$$\mathcal{D}_n = \{(X_i, S_i, Y_i)\}_{i=1}^n \overset{\text{i.i.d.}}{\sim} \mathbb{P}, \text{ and } \mathcal{D}_N = \{(X_i, S_i)\}_{i=n+1}^{n+N} \overset{\text{i.i.d.}}{\sim} \mathbb{P}_{(X,S)} \ ,$$

where $\mathbb{P}_{(X,S)}$ is the marginal distribution of the vector $(X, S)$. We additionally assume that the estimator $\hat{\eta}$ of the regression function is constructed based on $\mathcal{D}_n$, independently of $\mathcal{D}_N$. Let us denote by $\hat{\mathbb{E}}_{X|S=1}, \hat{\mathbb{E}}_{X|S=0}$ expectations taken *w.r.t.* the empirical distributions induced by $\mathcal{D}_N$, that is,

$$\hat{\mathbb{P}}_{X|S=s} = \frac{1}{|\{(X, S) \in \mathcal{D}_N \,:\, S = s\}|} \sum_{\{(X,S) \in \mathcal{D}_N \,:\, S=s\}} \delta_X \ ,$$

for all $s \in \{0, 1\}$, and by $\hat{\mathbb{E}}_S$ expectation taken *w.r.t.* the empirical measure of $S$, that is, $\hat{\mathbb{P}}_S = \frac{1}{N} \sum_{(X,S) \in \mathcal{D}_N} \delta_S$.

**Remark 3.1.** *In theory, the empirical distributions might be not well defined, since they are* only *valid if the unlabeled dataset $\mathcal{D}_N$ is composed of features from* both *groups. We show how to bypass this problem theoretically in supplementary material. Nevertheless, this remark has little to no impact in practice and in most situations these quantities are well defined.*

Based on the estimator $\hat{\eta}$ and the unlabeled sample $\mathcal{D}_N$, let us introduce the following estimators for each $s \in \{0, 1\}$

$$\hat{\mathbb{P}}(Y = 1, S = s) := \hat{\mathbb{E}}_{X|S=s}[\hat{\eta}(X, s)]\hat{\mathbb{P}}_S(S = s) \ .$$

Using the above estimators a straightforward procedure to mimic the optimal classifier $g^*$ provided by Proposition 2.3 is to employ a *plug-in rule* $\hat{g}$, obtained by replacing all the unknown quantities by either their empirical versions or their estimates. Specifically, we let $\hat{g}$ at $(x, s) \in \mathbb{R}^d \times \{0, 1\}$ as

$$\hat{g}(x, 1) = \mathbf{1}_{\left\{1 \leq \hat{\eta}(x,1)\left(2 - \frac{\hat{\theta}}{\hat{\mathbb{P}}(Y=1, S=1)}\right)\right\}}, \quad \hat{g}(x, 0) = \mathbf{1}_{\left\{1 \leq \hat{\eta}(x,0)\left(2 + \frac{\hat{\theta}}{\hat{\mathbb{P}}(Y=1, S=0)}\right)\right\}} \ . \tag{4}$$

It remains to define the value of $\hat{\theta}$, clearly it is desirable to mimic the condition that is satisfied by $\theta^*$ in Proposition 2.3. To this end, we make use of the unlabeled data $\mathcal{D}_N$ and of the estimator $\hat{\eta}$ previously built from the labeled dataset $\mathcal{D}_n$. Consequently, we define a data-driven version of unfairness $\Delta(g, \mathbb{P})$, which allows to construct an approximation $\hat{\theta}$ of the true value $\theta^*$.

**Definition 3.2** (Empirical unfairness). *For any classifier g, an estimator $\hat{\eta}$ based on $\mathcal{D}_n$, and unlabeled sample $\mathcal{D}_N$ the empirical unfairness is defined as*

$$\hat{\Delta}(g, \mathbb{P}) = \left| \frac{\hat{\mathbb{E}}_{X|S=1}\hat{\eta}(X,1)g(X,1)}{\hat{\mathbb{E}}_{X|S=1}\hat{\eta}(X,1)} - \frac{\hat{\mathbb{E}}_{X|S=0}\hat{\eta}(X,0)g(X,0)}{\hat{\mathbb{E}}_{X|S=0}\hat{\eta}(X,0)} \right| \ .$$

Notice that the empirical unfairness $\hat{\Delta}(g, \mathbb{P})$ is data-driven, that is, it does not involve unknown quantities. One might wonder why it is an empirical version of the quantity $\Delta(g, \mathbb{P})$ in Definition 2.4 and what is the reason to introduce it. The definition reveals itself when we rewrite the population of unfairness $\Delta(g, \mathbb{P})$ using[1] the identity

$$\mathbb{P}\left(g(X, S) = 1 \mid S = s, Y = 1\right) = \frac{\mathbb{P}(g(X,S)=1, Y=1 \mid S=s)}{\mathbb{P}(Y=1 \mid S=s)} = \frac{\mathbb{E}_{X \mid S=s}[\eta(X,s)g(X,s)]}{\mathbb{E}_{X \mid S=s}[\eta(X,s)]} \ .$$

Using the above expression we can rewrite

$$\Delta(g, \mathbb{P}) = \left| \frac{\mathbb{E}_{X \mid S=1}[\eta(X,1)g(X,1)]}{\mathbb{E}_{X \mid S=1}[\eta(X,1)]} - \frac{\mathbb{E}_{X \mid S=0}[\eta(X,0)g(X,0)]}{\mathbb{E}_{X \mid S=0}[\eta(X,0)]} \right| \ .$$

Hence, the passage from the population unfairness to its empirical version in Definition 3.2 formally reduces to substituting "hats" to all the unknown quantities.

Using Definition 3.2, a logical estimator $\hat{\theta}$ of $\theta^*$ can be obtained as

$$\hat{\theta} \in \underset{\theta \in [-2,2]}{\arg\min} \, \hat{\Delta}(\hat{g}_\theta, \mathbb{P}) \ ,$$

where, for all $\theta \in [-2, 2]$, $\hat{g}_\theta$ is defined at $(x, s) \in \mathbb{R}^d \times \{0, 1\}$ as

$$\hat{g}_\theta(x, 1) = \mathbf{1}_{\left\{ 1 \leq \hat{\eta}(x,1)\left( 2 - \frac{\theta}{\hat{\mathbb{P}}(Y=1, S=1)} \right) \right\}}, \quad \hat{g}_\theta(x, 0) = \mathbf{1}_{\left\{ 1 \leq \hat{\eta}(x,0)\left( 2 + \frac{\theta}{\hat{\mathbb{P}}(Y=1, S=0)} \right) \right\}} \ . \tag{5}$$

In this case, the algorithm $\hat{g}$ that we propose is such that $\hat{g} \equiv \hat{g}_{\hat{\theta}}$. It is crucial to mention that since the quantity $\hat{\Delta}(\hat{g}_\theta, \mathbb{P})$ is empirical, then there might be no $\theta$ which delivers zero for the empirical unfairness. This is exactly the reason we perform a minimization of this quantity.

**Remark 3.3.** *Even though we believe that the introduction of the unlabeled sample is one of the strong points of our approach, this sample may not be available on some benchmark datasets. In this case, we can simply randomly split the data into two parts disregarding labels in one of them, or alternatively we can use the same sample twice. The second path is not directly justified by our theoretical results, yet, let us suggest the following intuitive explanation for this approach. On the first and the second steps, our procedure approximates two* independent *parts of the distribution $\mathbb{P}$ of the random tuple $(X, S, Y)$. Indeed, following the factorization $\mathbb{P} = \mathbb{P}_{Y \mid X, S} \otimes \mathbb{P}_{(X, S)}$, the first step of our procedure approximates $\mathbb{P}_{Y \mid X, S}$, whereas the second step is aimed at $\mathbb{P}_{(X, S)}$ which is independent from $\mathbb{P}_{Y \mid X, S}$. In our experiments, reported in Section 5, we exploited the same set of data for both $\mathcal{D}_n$ and $\mathcal{D}_N$, since no unlabelled sample were available and splitting the dataset would have reduced the quality of the trained model because the datasets have a small sample size.*

## 4 Consistency

In this section we establish that the proposed procedure is consistent. To present our theoretical results we impose two assumptions on the estimator $\hat{\eta}$ and demonstrate how to satisfy them in practice.

**Assumption 4.1.** *The estimator $\hat{\eta}$ which is constructed on $\mathcal{D}_n$ satisfies for all $s \in \{0, 1\}$*

*(i)* $\mathbb{E}_{\mathcal{D}_n} \mathbb{E}_{X \mid S=s} |\eta(X, S) - \hat{\eta}(X, S)| \to 0$ *as $n \to \infty$;*

*(ii) There exists a sequence $c_{n,N} > 0$ satisfying $\frac{1}{c_{n,N} \sqrt{N}} = o_{n,N}(1)$ and $c_{n,N} = o_{n,N}(1)$ such that $\mathbb{E}_{X \mid S=s}[\hat{\eta}(X, S)] \geq c_{n,N}$ almost surely.*

**Remark 4.2.** *There are two parts in Assumption 4.1, the first one requires a consistent estimator in $\ell_1$ norm. This first assumption is rather weak, since there are many different available consistent estimators for the regression function in the literature, including the Maximum likelihood estimator [45] for Gaussian Generative Model, local polynomial estimator [4] for $\beta$-Hölder smooth regression function $\eta(\cdot, s)$, regularized logistic regression [42] for Generalized Linear Model, $k$-Nearest Neighbors estimator [16] for Lipschitz regression function $\eta(\cdot, s)$, and random forest type estimators in various settings [3, 7, 21, 41].*
*The second part of Assumption 4.1 means that $\mathbb{E}_{X \mid S=s}[\hat{\eta}(X, s)]$ is lower bounded by a positive term*

*vanishing as $N, n$ grow to infinity. This condition can be introduced artificially to any predefined estimator. Indeed, assume that we have a consistent estimator $\tilde{\eta}$ and let $\hat{\eta}(x, s) = \max\{\tilde{\eta}(x, s), c_{n,N}\}$, then the second item of the assumption is satisfied in even a stronger form. Moreover, this estimator $\hat{\eta}$ remains consistent, since using the triangle inequality and the fact that $|\hat{\eta}(x, s) - \tilde{\eta}(x, s)| \leq c_{n,N}$ for all $x \in \mathbb{R}^d$, we have*

$$\mathbb{E}_{\mathcal{D}_n}\mathbb{E}_{X|S=s}\,|\eta(X, s) - \hat{\eta}(X, s)| \leq \mathbb{E}_{\mathcal{D}_n}\mathbb{E}_{X|S=s}\,|\eta(X, s) - \tilde{\eta}(X, s)| + c_{n,N} \to 0 \ .$$

Additionally, we impose one more condition on the estimator $\hat{\eta}$ that was already successfully used in the context of confidence set classification [11, 15].

**Assumption 4.3.** *The estimator $\hat{\eta}$ is such that for all $s \in \{0, 1\}$ the mapping*

$$t \mapsto \mathbb{P}\left(\hat{\eta}(X, s) \leq t \,|\, S = s\right) \ ,$$

*is continuous on $(0, 1)$ almost surely.*

In our settings this assumption allows us to show that the value of $\hat{\Delta}(\hat{g}, \mathbb{P})$ cannot be large, that is, the empirical unfairness of the proposed procedure is small or zero. As we shall see, a control on the empirical unfairness $\hat{\Delta}(\hat{g}, \mathbb{P})$ in Definition 3.2 is crucial in proving that the proposed procedure $\hat{g}$ achieves both asymptotic fairness and risk consistency.

**Remark 4.4.** *Assumption 4.3 is equivalent to say that there are no atoms in the estimated regression function. It can be fulfilled by a simple modification of any preliminary estimator, by adding a* small *deterministic "noise", the amplitude of which must be decreasing with $n, N$ in order to preserve statistical consistency.*

Our remarks suggest that both Assumptions 4.1 and 4.3 can be easily satisfied in a variety of practical settings and the most demanding part of these assumptions is the consistency of $\hat{\eta}$.

The next result establishes the statistical consistency of the proposed algorithm.

**Theorem 4.5** (Asymptotic properties)**.** *Under Assumptions 2.2, 4.1, and 4.3 the proposed algorithm satisfies*

$$\lim_{n,N \to \infty} \mathbb{E}_{(\mathcal{D}_n, \mathcal{D}_N)}[\Delta(\hat{g}, \mathbb{P})] = 0 \quad \text{and} \quad \lim_{n,N \to \infty} \mathbb{E}_{(\mathcal{D}_n, \mathcal{D}_N)}[\mathcal{R}(\hat{g})] \leq \mathcal{R}(g^*) \ .$$

*Proof sketch.* In order to establish statistical consistency of the proposed procedure, we follow the strategy of [11, 15], that is, we first introduce an intermediate pseudo-estimator $\tilde{g}$ as follows

$$\tilde{g}(x, 1) = \mathbf{1}_{\left\{1 \leq \hat{\eta}(x,1)\left(2 - \frac{\tilde{\theta}}{\mathbb{E}_{X|S=1}[\hat{\eta}(X,1)]\mathbb{P}(S=1)}\right)\right\}}, \quad \tilde{g}(x, 0) = \mathbf{1}_{\left\{1 \leq \hat{\eta}(x,0)\left(2 + \frac{\tilde{\theta}}{\mathbb{E}_{X|S=0}[\hat{\eta}(X,0)]\mathbb{P}(S=0)}\right)\right\}}, \quad (6)$$

where $\tilde{\theta}$ is chosen such that

$$\frac{\mathbb{E}_{X|S=1}\left[\hat{\eta}(X, 1)\tilde{g}(X, 1)\right]}{\mathbb{E}_{X|S=1}[\hat{\eta}(X, 1)]} = \frac{\mathbb{E}_{X|S=0}\left[\hat{\eta}(X, 0)\tilde{g}(X, 0)\right]}{\mathbb{E}_{X|S=0}[\hat{\eta}(X, 0)]} \ . \quad (7)$$

Note that by Assumption 4.3 such a value $\tilde{\theta}$ always exists. Intuitively, the classifier $\tilde{g}$ "*knows*" the marginal distribution of $(X, S)$, that is, it knows both $\mathbb{P}_{X|S}$ and $\mathbb{P}_S$. It is seen as an idealized version of $\hat{g}$, where the uncertainty is only induced by the lack of knowledge of the regression function $\eta$.

We express the excess risk as a sum of two terms, $\mathbb{E}_{\mathcal{D}_n}[\mathcal{R}(\tilde{g})] - \mathcal{R}(g^*) + \mathbb{E}_{(\mathcal{D}_n, \mathcal{D}_N)}[\mathcal{R}(\hat{g}) - \mathcal{R}(\tilde{g})]$. We show that the first can be bounded by the $\ell_1$ distance between $\hat{\eta}$ and $\eta$, and thanks to the consistency of $\hat{\eta}$ it does converge to zero. The handling of the second term is move involved, but we are able to show that it reduces to a study of suprema of empirical processes conditionally on the labeled sample $\mathcal{D}_n$.

To demonstrate that the proposed algorithm is asymptotically fair, we first show that

$$\mathbb{E}_{(\mathcal{D}_n, \mathcal{D}_N)}[\Delta(\hat{g}, \mathbb{P})] \leq \mathbb{E}_{(\mathcal{D}_n, \mathcal{D}_N)}[\hat{\Delta}(\hat{g}, \mathbb{P})] + o_{n,N}(1) \ .$$

At last, the continuity Assumption 4.3 alongside with means of theory of empirical processes allow to demonstrate that the term $\mathbb{E}_{(\mathcal{D}_n, \mathcal{D}_N)}[\hat{\Delta}(\hat{g}, \mathbb{P})]$ converges to zero when $N$ growth. $\qquad\square$

**Remark 4.6.** *Let us mention that it is possible to present our result in a finite sample regime, since our proof of consistency is based on non-asymptotic theory of empirical processes. However, the actual rate of convergence depends on the rate of $\ell_1$-norm estimation of the regression function $\eta$, which can vary significantly from one setup to another. That is why we decided to present our result in the asymptotic sense.*

| Method | Arrhythmia | | COMPAS | | Adult | | German | | Drug | |
|---|---|---|---|---|---|---|---|---|---|---|
| | ACC | DEO | ACC | DEO | ACC | DEO | ACC | DEO | ACC | DEO |
| Lin.SVM | 0.78±0.07 | 0.13±0.04 | 0.75±0.01 | 0.15±0.02 | 0.80 | 0.13 | 0.69±0.04 | 0.11±0.10 | 0.81±0.02 | 0.41±0.06 |
| Lin.LR | 0.79±0.06 | 0.13±0.05 | 0.76±0.02 | 0.16±0.02 | 0.81 | 0.12 | 0.67±0.05 | 0.12±0.11 | 0.80±0.01 | 0.42±0.05 |
| Lin.SVM+Hardt | 0.74±0.06 | 0.07±0.04 | 0.67±0.03 | 0.21±0.09 | 0.80 | 0.10 | 0.61±0.15 | 0.15±0.13 | 0.77±0.02 | 0.22±0.09 |
| Lin.LR+Hardt | 0.75±0.04 | 0.08±0.05 | 0.67±0.02 | 0.18±0.07 | 0.81 | 0.09 | 0.62±0.05 | 0.13±0.09 | 0.76±0.01 | 0.18±0.04 |
| Zafar | 0.71±0.03 | 0.03±0.02 | 0.69±0.02 | 0.10±0.06 | 0.78 | 0.05 | 0.62±0.09 | 0.13±0.11 | 0.69±0.03 | 0.02±0.07 |
| Lin.Donini | 0.79±0.07 | 0.04±0.03 | 0.76±0.01 | 0.04±0.03 | 0.77 | 0.01 | 0.69±0.04 | 0.05±0.03 | 0.79±0.02 | 0.05±0.03 |
| Lin.SVM+Ours | 0.75±0.08 | 0.04±0.04 | 0.73±0.01 | 0.05±0.02 | 0.79 | 0.03 | 0.68±0.04 | 0.04±0.03 | 0.78±0.02 | 0.01±0.02 |
| Lin.LR+Ours | 0.75±0.06 | 0.04±0.05 | 0.74±0.02 | 0.06±0.02 | 0.80 | 0.03 | 0.67±0.05 | 0.04±0.03 | 0.77±0.03 | 0.02±0.02 |
| SVM | 0.78±0.06 | 0.13±0.04 | 0.73±0.01 | 0.14±0.02 | 0.82 | 0.14 | 0.74±0.03 | 0.10±0.06 | 0.81±0.04 | 0.38±0.03 |
| LR | 0.79±0.05 | 0.12±0.04 | 0.74±0.01 | 0.14±0.02 | 0.81 | 0.15 | 0.75±0.03 | 0.11±0.06 | 0.82±0.01 | 0.37±0.03 |
| RF | 0.83±0.03 | 0.09±0.02 | 0.77±0.02 | 0.11±0.02 | 0.86 | 0.12 | 0.78±0.02 | 0.09±0.04 | 0.86±0.01 | 0.29±0.02 |
| SVM+Hardt | 0.74±0.06 | 0.07±0.04 | 0.71±0.02 | 0.08±0.02 | 0.82 | 0.11 | 0.71±0.03 | 0.11±0.18 | 0.75±0.11 | 0.14±0.08 |
| LR+Hardt | 0.73±0.05 | 0.10±0.04 | 0.70±0.02 | 0.09±0.02 | 0.80 | 0.12 | 0.72±0.04 | 0.09±0.06 | 0.77±0.03 | 0.11±0.04 |
| RF+Hardt | 0.79±0.03 | 0.07±0.01 | 0.76±0.01 | 0.07±0.02 | 0.83 | 0.05 | 0.76±0.02 | 0.06±0.04 | 0.82±0.01 | 0.09±0.02 |
| Donini | 0.79±0.09 | 0.03±0.02 | 0.73±0.01 | 0.05±0.03 | 0.81 | 0.01 | 0.73±0.04 | 0.05±0.03 | 0.80±0.03 | 0.07±0.05 |
| SVM+Ours | 0.77±0.07 | 0.04±0.02 | 0.72±0.02 | 0.06±0.02 | 0.80 | 0.02 | 0.73±0.03 | 0.04±0.06 | 0.79±0.02 | 0.05±0.01 |
| LR+Ours | 0.77±0.06 | 0.04±0.02 | 0.73±0.01 | 0.06±0.02 | 0.80 | 0.02 | 0.73±0.02 | 0.04±0.06 | 0.80±0.01 | 0.05±0.02 |
| RF+Ours | 0.81±0.04 | 0.03±0.01 | 0.76±0.02 | 0.04±0.02 | 0.85 | 0.03 | 0.77±0.02 | 0.02±0.02 | 0.83±0.01 | 0.04±0.02 |

Table 1: Results (average ± standard deviation, when a fixed test set is not provided) for all the datasets, concerning ACC and DEO.

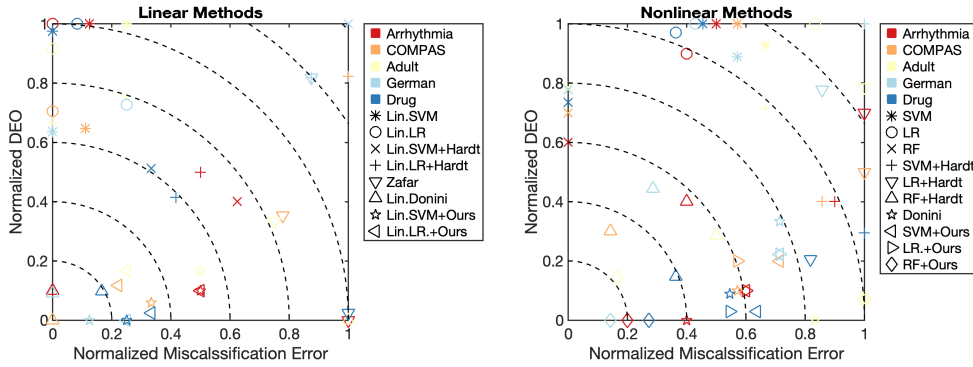

Figure 1: Results of Table 1 of linear (left) and nonlinear (right) methods when the error and the DEO are normalized in $[0, 1]$ column-wise. Different colors and symbols refer to different datasets and method respectively. The closer a point is to the origin, the better the result is.

# 5  Experimental results

In this section, we present numerical experiments with the proposed method. The source code we used to perform the experiments can be found at `https://github.com/lucaoneto/NIPS2019_Fairness`.

We follow the protocol outlined in [17]. We consider the following datasets: Arrhythmia, COMPAS, Adult, German, and Drug[2] and compare the following algorithms: Linear Support Vector Machines (Lin.SVM), Support Vector Machines with the Gaussian kernel (SVM), Linear Logistic Regression (Lin.LR), Logistic Regression with the Gaussian kernel (LR), Hardt method [22] to all approaches (Hardt), Zafar method [48] implemented with the code provided by the authors for the linear case[3], the Linear (Lin.Donini) and the Non Linear methods (Donini) proposed in [17] and freely available[4], and also Random Forests (RF). Then, since Lin.SVM, SVM, Lin.LR, LR, and RF have also the possibility to output a probability together with the classification, we applied our method in all these cases.

In all experiments, we collect statistics concerning the classification accuracy (ACC), namely probability to correctly classify a sample, and the Difference of Equal Opportunity (DEO) in Definition 2.1. For Arrhythmia, COMPAS, German and Drug datasets we split the data in two parts (70%

| RF+Ours | COMPAS | | Adult | |
|---|---|---|---|---|
| | ACC | DEO | ACC | DEO |
| $\mathcal{D}_n = {}^1/_{10}$ | $0.68 \pm 0.03$ | $0.07 \pm 0.02$ | $0.79 \pm 0.02$ | $0.06 \pm 0.02$ |
| $\mathcal{D}_n = {}^1/_{10}, \mathcal{D}_N = {}^1/_{10}$ | $0.68 \pm 0.03$ | $0.07 \pm 0.02$ | $0.79 \pm 0.02$ | $0.06 \pm 0.02$ |
| $\mathcal{D}_n = {}^1/_{10}, \mathcal{D}_N = {}^2/_{10}$ | $0.68 \pm 0.03$ | $0.07 \pm 0.02$ | $0.79 \pm 0.02$ | $0.06 \pm 0.02$ |
| $\mathcal{D}_n = {}^1/_{10}, \mathcal{D}_N = {}^4/_{10}$ | $0.70 \pm 0.02$ | $0.06 \pm 0.02$ | $0.79 \pm 0.02$ | $0.05 \pm 0.01$ |
| $\mathcal{D}_n = {}^1/_{10}, \mathcal{D}_N = {}^8/_{10}$ | $0.71 \pm 0.02$ | $0.05 \pm 0.01$ | $0.80 \pm 0.02$ | $0.04 \pm 0.01$ |

Table 2: Impact of the size of the unlabeled dataset on ACC and DEO. The size of the labeled sample $\mathcal{D}_n$ is fixed to $1/10$ of the original dataset. The unlabeled $\mathcal{D}_N$ is initially empty (as in previous experiments of Table 1), and then increases from $1/10$ to $8/10$ of the original dataset.

train and 30% test), this procedure is repeated 30 times, and we reported the average performance on the test set alongside its standard deviation. For the Adult dataset, we used the provided split of train and test sets. Unless otherwise stated, we employ two steps in the 10-fold CV procedure proposed in [17] to select the best hyperparameters with the training set[5]. In the first step, the value of the hyperparameters with the highest accuracy is identified. In the second step, we shortlist all the hyperparameters with accuracy close to the best one (in our case, above $90\%$ of the best accuracy). Finally, from this list, we select the hyperparameters with the lowest DEO.

We also present in Figure 1 the results of Table 1 for linear (left) and nonlinear (right) methods, when the error (one minus ACC) and the DEO are normalized in $[0, 1]$ column-wise. In the figure, different colors and symbols refer to different datasets and methods, respectively. The closer a point is to the origin, the better the result is.

From Table 1 and Figure 1 it is possible to observe that the proposed method outperforms all methods except the one of [17] for which we obtain comparable performance. Nevertheless, note that our method is more general than the one of [17], since it can be applied to any algorithms which return a probability estimator (better if consistent since this will allow us to have a full consistent approach also from the fairness point of view). In fact, on these datasets, RF, which cannot be made trivially fair with the approach proposed in [17], outperforms all the available methods.

Note that the results reported in Table 1 differ from the one reported in [17] since the proposed method requires the knowledge of the sensitive variable at classification time, so Table 1 reports just this case. That is, the functional form of the model explicitly depends on the sensitive variable $s \in \{0, 1\}$. Many authors, point out that this may not be permitted in several practical scenarios (see e.g. [19, 39] and reference therein). Yet, removing the sensitive variable from the functional form of the model does not ensure that the sensitive variable is not considered by the model itself. We refer to [36] for the in-depth discussion on this issue. Further, the method in [22] explicitly requires the knowledge of the sensitive variable for their thresholding procedure. In Appendix E we show how to modify our method in order to derive a fair optimal classifier without the sensitive variable $s$ in the functional form of the model. Moreover, we propose a modification of our approach which does not use $s$ at decision time and perform additional numerical comparison in this context. We arrive to similar conclusions about the performance of our method as in this section. Yet, the consistency results are not available for this methods and are left for future investigation.

In Table 2 we demonstrate the impact of the unlabeled data size on the performance of the proposed algorithm. Since the above benchmark datasets are not provide with additional unlabeled data, we deploy the following data generation procedure: we randomly select $1/10$ observations in each dataset and assign it to the labeled sample $\mathcal{D}_n$; consequently, the size of the unlabeled sample $\mathcal{D}_N$ increases from 0 to $8/10$ samples that were not assigned to the labeled sample $\mathcal{D}_n$. This data generation procedure is applied to COMPAS and Adult datasets. Finally, we apply our method using the random forest algorithm using the cross-validation scheme employed in the previous experiments. The above above pipeline is repeated 30 times and the variance of the results is reported in Table 2. We can see that both DEO and ACC are improving with $N$, highlighting the importance of the unlabeled data. We believe that the improvement could have been more significant if the unlabeled data were provided initially.

# 6 Conclusion

Using the notion of equal opportunity we have derived a form of the fair optimal classifier based on group-dependent threshold. Relying on this result we have proposed a semi-supervised plug-in method which enjoys strong theoretical guarantees under mild assumptions. Importantly, our algorithm can be implemented on top of any base classifier which has conditional probabilities as outputs. We have conducted an extensive numerical evaluation comparing our procedure against the state-of-the-art approaches and have demonstrated that our procedure performs well in practice. In future works we would like to extend our analysis to other fairness measures as well as provide consistency results for the algorithm which does not use the sensitive feature at the decision time. Finally, we note that our consistency result is constructive and could be used to derive non-asymptotic rates of convergence for the proposed method, relying upon available rates for the regression function estimator.

**Acknowledgments**

This work was supported in part by SAP SE, by the Amazon AWS Machine Learning Research Award, by CISCO, and by the Labex Bézout of Université Paris-Est.

## Footnotes

[1]Note additionally that for all $s \in \{0, 1\}$ we can write $\mathbf{1}_{\{Y=1, g(X,s)=1\}} \equiv Y g(X, s)$, since both $Y$ and $g$ are binary.

[2]For more information about these datasets please refer to [17].

[3]Python code for [48]: `https://github.com/mbilalzafar/fair-classification`

[4]Python code for [17]: `https://github.com/jmikko/fair_ERM`

[5]The regularization parameter (for all method) and the RBF kernel with 30 values, equally spaced in logarithmic scale between $10^{-4}$ and $10^4$. For RF the number of trees has been set to 1000 and the size of the subset of features optimized at each node has been search in $\{d, \lceil d^{15/16} \rceil, \lceil d^{7/8} \rceil, \lceil d^{3/4} \rceil, \lceil d^{1/2} \rceil, \lceil d^{1/4} \rceil, \lceil d^{1/8} \rceil, \lceil d^{1/16} \rceil, 1\}$ where $d$ is the number of features in the dataset.

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
