[Supplementary Material]

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

# Supplementary material for "Leveraging Labeled and Unlabeled Data for Consistent Fair Binary Classification"

## A Optimal classifier

*Proof of Proposition 2.3.* Let us study the following minimization problem

$$(*) := \min_{g \in \mathcal{G}} \{\mathcal{R}(g) : \mathbb{P}(g(X,S){=}1 \,|\, Y{=}1, S{=}1) = \mathbb{P}(g(X,S){=}1 \,|\, Y{=}1, S{=}0)\} \ .$$

Using the weak duality we can write

$$(*) = \min_{g \in \mathcal{G}} \max_{\lambda \in \mathbb{R}} \{\mathcal{R}(g) + \lambda (\mathbb{P}(g(X,S){=}1 \,|\, Y{=}1, S{=}1) - \mathbb{P}(g(X,S){=}1 \,|\, Y{=}1, S{=}0))\}$$

$$\geq \max_{\lambda \in \mathbb{R}} \min_{g \in \mathcal{G}} \{\mathcal{R}(g) + \lambda (\mathbb{P}(g(X,S){=}1 \,|\, Y{=}1, S{=}1) - \mathbb{P}(g(X,S){=}1 \,|\, Y{=}1, S{=}0))\} =: (**) \ .$$

We first study the objective function of the max min problem $(**)$, which is equal to

$$\mathbb{P}(g(X,S) \neq Y) + \lambda (\mathbb{P}(g(X,S){=}1 \,|\, Y{=}1, S{=}1) - \mathbb{P}(g(X,S){=}1 \,|\, Y{=}1, S{=}0)) \ .$$

The first step of the proof is to simplify the expression above to linear functional of the classifier $g$. Notice that we can write for the first term

$$
\begin{aligned}
\mathbb{P}(g(X,S) \neq Y) &= \mathbb{P}(g(X,S){=}0, Y{=}1) + \mathbb{P}(g(X,S){=}1, Y{=}0) \\
&= \mathbb{P}(g(X,S){=}1) + \mathbb{P}(Y{=}1) - \mathbb{P}(g(X,S){=}1, Y{=}1) - \mathbb{P}(g(X,S){=}1, Y{=}1) \\
&= \mathbb{P}(g(X,S){=}1) + \mathbb{P}(Y{=}1) - 2\mathbb{P}(g(X,S){=}1, Y{=}1) \\
&= \mathbb{P}(Y{=}1) + \mathbb{E}[g(X,S)] - 2\mathbb{E}\left[\mathbf{1}_{\{g(X,S){=}1, Y{=}1\}} \,|\, S{=}1\right] \mathbb{P}(S{=}1) \\
&\quad - 2\mathbb{E}\left[\mathbf{1}_{\{g(X,S){=}1, Y{=}1\}} \,|\, S{=}0\right] \mathbb{P}(S{=}0) \\
&= \mathbb{P}(Y{=}1) + \mathbb{E}[g(X,S)] - 2\mathbb{E}_{X|S=1}[g(X,1)\eta(X,1)]\mathbb{P}(S{=}1) \\
&\quad - 2\mathbb{E}_{X|S=0}[g(X,0)\eta(X,0)]\mathbb{P}(S{=}0) \\
&= \mathbb{P}(Y{=}1) - \mathbb{E}_{X|S=1}[g(X,1)(2\eta(X,1) - 1)]\mathbb{P}(S{=}1) \\
&\quad - \mathbb{E}_{X|S=0}[g(X,0)(2\eta(X,0) - 1)]\mathbb{P}(S{=}0) \ ,
\end{aligned}
$$

moreover, we can write for the rest

$$\mathbb{P}(g(X,S) = 1 \,|\, Y = 1, S = 1) = \frac{\mathbb{P}(g(X,S) = 1, Y = 1 \,|\, S = 1)}{\mathbb{P}(Y = 1 \,|\, S = 1)} = \frac{\mathbb{E}_{X|S=1}[g(X,1)\eta(X,1)]}{\mathbb{P}(Y = 1 \,|\, S = 1)} \ ,$$

$$\mathbb{P}(g(X,S) = 1 \,|\, Y = 1, S = 0) = \frac{\mathbb{P}(g(X,S) = 1, Y = 1 \,|\, S = 0)}{\mathbb{P}(Y = 1 \,|\, S = 0)} = \frac{\mathbb{E}_{X|S=0}[g(X,0)\eta(X,0)]}{\mathbb{P}(Y = 1 \,|\, S = 0)} \ .$$

Using these, the objective of $(**)$ can be simplified as

$$
\begin{aligned}
\mathbb{P}(Y = 1) &+ \mathbb{E}_{X|S=1}\left[g(X,1)\left(\eta(X,1)\left(\frac{\lambda}{\mathbb{P}(Y = 1 \,|\, S = 1)} - 2\mathbb{P}(S = 1)\right) + \mathbb{P}(S = 1)\right)\right] \\
&+ \mathbb{E}_{X|S=0}\left[g(X,0)\left(\eta(X,0)\left(-\frac{\lambda}{\mathbb{P}(Y = 1 \,|\, S = 0)} - 2\mathbb{P}(S = 0)\right) + \mathbb{P}(S = 0)\right)\right] \ .
\end{aligned}
$$

Clearly, for every $\lambda \in \mathbb{R}$ a minimizer $g_\lambda^*$ of the problem $(**)$ can be written for all $x \in \mathbb{R}^d$ as

$$g_\lambda^*(x,1) = \mathbf{1}_{\left\{\eta(X,1)\left(\frac{\lambda}{\mathbb{P}(Y=1 \,|\, S=1)} - 2\mathbb{P}(S=1)\right) + \mathbb{P}(S=1) \leq 0\right\}} = \mathbf{1}_{\left\{1 - \eta(X,1)\left(2 - \frac{\lambda}{\mathbb{P}(Y=1, S=1)}\right) \leq 0\right\}}$$

$$g_\lambda^*(x,0) = \mathbf{1}_{\left\{\eta(X,0)\left(-\frac{\lambda}{\mathbb{P}(Y=1 \,|\, S=0)} - 2\mathbb{P}(S=0)\right) + \mathbb{P}(S=0) \leq 0\right\}} = \mathbf{1}_{\left\{1 - \eta(X,0)\left(2 + \frac{\lambda}{\mathbb{P}(Y=1, S=0)}\right) \leq 0\right\}} \ .$$

At this moment it is interesting to reflect on this result. Indeed, for $\lambda = 0$ we recover the classical optimal predictor in the context of binary classification. Substituting this classifier into the objective of $(**)$ we arrive at

$$
\begin{aligned}
(**) = \mathbb{P}(Y = 1) - \min_{\lambda \in \mathbb{R}} \bigg\{ &\mathbb{E}_{X|S=1}\left(\eta(X,1)\left(2\mathbb{P}(S = 1) - \frac{\lambda}{\mathbb{P}(Y = 1 \,|\, S = 1)}\right) - \mathbb{P}(S = 1)\right)_+ \\
&+ \mathbb{E}_{X|S=0}\left(\eta(X,0)\left(2\mathbb{P}(S = 0) + \frac{\lambda}{\mathbb{P}(Y = 1 \,|\, S = 0)}\right) - \mathbb{P}(S = 0)\right)_+ \bigg\} \ .
\end{aligned}
$$

It is important to observe that the mappings

$$\lambda \mapsto \mathbb{E}_{X|S=1}\left(\eta(X,1)\left(2\mathbb{P}(S=1) - \frac{\lambda}{\mathbb{P}\left(Y=1\,|\,S=1\right)}\right) - \mathbb{P}(S=1)\right)_{+}$$

$$\lambda \mapsto \mathbb{E}_{X|S=0}\left(\eta(X,0)\left(2\mathbb{P}(S=0) + \frac{\lambda}{\mathbb{P}\left(Y=1\,|\,S=0\right)}\right) - \mathbb{P}(S=0)\right)_{+} ,$$

are convex, therefore we can write the first order optimality conditions as

$$0 \in \partial_{\lambda}\mathbb{E}_{X|S=1}\left(\eta(X,1)\left(2\mathbb{P}(S=1) - \frac{\lambda}{\mathbb{P}\left(Y=1\,|\,S=1\right)}\right) - \mathbb{P}(S=1)\right)_{+}$$

$$+ \partial_{\lambda}\mathbb{E}_{X|S=0}\left(\eta(X,0)\left(2\mathbb{P}(S=0) + \frac{\lambda}{\mathbb{P}\left(Y=1\,|\,S=0\right)}\right) - \mathbb{P}(S=0)\right)_{+} .$$

Clearly, under Assumption 2.2 this subgradient is reduced to the gradient almost surely, thus we have the following condition on the optimal value of $\lambda^{*}$

$$\frac{\mathbb{E}_{X|S=1}\left[\eta(X,1)g_{\lambda^*}^{*}(X,1)\right]}{\mathbb{P}\left(Y=1\,|\,S=1\right)} = \frac{\mathbb{E}_{X|S=0}\left[\eta(X,0)g_{\lambda^*}^{*}(X,0)\right]}{\mathbb{P}\left(Y=1\,|\,S=0\right)} ,$$

and the pair $(\lambda^{*}, g_{\lambda^*}^{*})$ is a solution of the dual problem $(**)$. Notice that the previous condition can be written as

$$\mathbb{P}\left(g_{\lambda^*}^{*}(X,S)=1\,|\,Y=1,S=1\right) = \mathbb{P}\left(g_{\lambda^*}^{*}(X,S)=1\,|\,Y=1,S=0\right) .$$

This implies that the classifier $g_{\lambda^*}^{*}$ is fair, that is, it satisfies Definition 2.1. Finally, it remains to show that $g_{\lambda^*}^{*}$ is actually an optimal classifier, indeed, since $g_{\lambda^*}^{*}$ is fair we can write on the one hand

$$\mathcal{R}(g_{\lambda^*}^{*}) \geq \min_{g \in \mathcal{G}}\left\{\mathcal{R}(g)\,:\,\mathbb{P}\left(g(X,S)=1\,|\,Y=1,S=1\right) = \mathbb{P}\left(g(X,S)=1\,|\,Y=1,S=0\right)\right\} = (*).$$

On the other hand the pair $(\lambda^{*}, g_{\lambda^*}^{*})$ is a solution of the dual problem $(**)$, thus we have

$$(*) \geq \mathcal{R}(g_{\lambda^*}^{*}) + \lambda^{*}\left(\mathbb{P}\left(g_{\lambda^*}^{*}(X,S)=1\,|\,Y=1,S=1\right) - \mathbb{P}\left(g_{\lambda^*}^{*}(X,S)=1\,|\,Y=1,S=0\right)\right)$$
$$= \mathcal{R}(g_{\lambda^*}^{*}) .$$

It implies that the classifier $g_{\lambda^*}^{*}$ is optimal, hence $g^{*} \equiv g_{\lambda^*}^{*}$.

Finally, assume that $(2 - \theta^{*}/\mathbb{P}(Y=1,S=1)) \leq 0$, then, clearly $(2 + \theta^{*}/\mathbb{P}(Y=1,S=0)) > 0$, therefore, the condition on $\theta^{*}$ reads as

$$0 = \mathbb{E}_{X|S=0}\left[\eta(X,0)\mathbf{1}_{\left\{1 \leq \eta(X,0)\left(2 + \frac{\theta^*}{\mathbb{P}(Y=1,S=0)}\right)\right\}}\right] \geq \frac{\mathbb{P}\left(\eta(X,0) \geq \frac{1}{\left(2 + \frac{\theta^*}{\mathbb{P}(Y=1,S=0)}\right)}\,\Big|\,S=0\right)}{\left(2 + \frac{\theta^*}{\mathbb{P}(Y=1,S=0)}\right)}$$

$$\geq \frac{\mathbb{P}\left(\eta(X,0) \geq 1/2\,|\,S=0\right)}{\left(2 + \frac{\theta^*}{\mathbb{P}(Y=1,S=0)}\right)} > 0 ,$$

where the last inequality is due to Assumption 2.2. We arrive to contradiction, therefore $(2 - \theta^{*}/\mathbb{P}(Y=1,S=1)) > 0$. Similarly, we show that $(2 + \theta^{*}/\mathbb{P}(Y=1,S=0)) > 0$. Combination of both inequalities and the fact that for all $s \in \{0,1\}$ we have $\mathbb{P}(Y=1,S=s) \leq 1$ implies that $|\theta^{*}| \leq 2$. $\qquad\square$

## B   Auxiliary results

Before proceeding to the proof of our main result in Theorem 4.5, let us first introduce several auxiliary results. We suggest the reader to first understand these results omitting its proofs before proceeding further. We will use $C > 0$ as a generic constant which actually could be different from line to line, yet, this constant is always independent from $n, N$.

**Remark B.1.** *In our work it is assumed that the unlabeled dataset is sampled* i.i.d. *from* $\mathbb{P}_{(X,S)}$*, it implies that in theory this dataset could be composed of* only *features belonging to either of the group. Clearly, since* $\mathbb{P}(S = 1) > 0$ *and* $\mathbb{P}(S = 0) > 0$ *then this situation has an extremely small probability of appearing, in terms of* $N$*. There are various ways to alleviate this issue. The first one is conditioning on the event that we have at least one sample from each group, however, we have found that this approach unnecessarily over complicates our derivations and does not bring any insights. That is why, we follows another path, which is much simpler, though, might look a little strange at first sight. We actually augment* $\mathcal{D}_N$ *by* four *points* $(X_1, 1), (X_2, 1), (X_3, 0), (X_4, 0)$ *which are sampled as* $X_1, X_2 \overset{\text{i.i.d.}}{\sim} \mathbb{P}_{X|S=1}$ *and* $X_3, X_4 \overset{\text{i.i.d.}}{\sim} \mathbb{P}_{X|S=0}$*. Once it is done we can safely assume that* $\mathcal{D}_N$ *consists of* at least *two features from* each *group. The above is simply a technicality which allows to present our result in a correct way.*

The next lemma can be found in [13].

**Lemma B.2.** *Let* $Z$ *be a binomial random variable with parameters* $N, p$*, then for every* $\alpha \in \mathbb{R}$

$$\mathbb{E}[(1 + Z)^{\alpha}] = \mathcal{O}\left((Np)^{\alpha}\right) \ .$$

**Lemma B.3.** *For any classifier* $g$ *we have*

$$\mathcal{R}(g) = \mathbb{E}_{(X,S)}[\eta(X, S)] - \mathbb{E}_{(X,S)}[(2\eta(X, S) - 1)g(X, S)] \ .$$

*Proof.* We can write

$$
\begin{aligned}
\mathcal{R}(g) &:= \mathbb{P}(Y \neq g(X, S)) = \mathbb{E}[Y(1 - g(X, S))] + \mathbb{E}[(1 - Y)g(X, S)] \\
&= \mathbb{E}_{(X,S)}\eta(X, S)(1 - g(X, S)) + \mathbb{E}_{(X,S)}(1 - \eta(X, S))g(X, S) \\
&= \mathbb{E}_{(X,S)}[\eta(X, S)] - \mathbb{E}_{(X,S)}[(2\eta(X, S) - 1)g(X, S)] \ .
\end{aligned}
$$

$\square$

In what follows we shall often use the relations:

$$
\begin{aligned}
\mathbb{P}(Y = 1, S = s) &= \mathbb{P}(Y = 1 \mid S = s)\,\mathbb{P}(S = s) \ , \\
\mathbb{P}(Y = 1 \mid S = s) &= \mathbb{E}_{X|S=s}[\eta(X, s)] \ .
\end{aligned}
$$

which holds for all $s \in \{0, 1\}$.

## C   Proof of Theorem 4.5

Below we gather extra tools which are directly related to the proof of our main result, proof are provided in Appendix D. First lemma gives an upper on the quantity of unfairness $\Delta(g, \mathbb{P})$ in terms of its empirical version in Definition 3.2.

**Lemma C.1.** *Let* $g$ *be* any *classifier (data depended or not) and* $\hat{\eta}$ *be an estimator of the regression function* $\eta$ *constructed on* $\mathcal{D}_n$*. Then, almost surely we have*

$$\Delta(g, \mathbb{P}) \leq \hat{\Delta}(g, \mathbb{P}) + 2\underbrace{\frac{\mathbb{E}_{X|S=1}\left|\eta(X, 1) - \hat{\eta}(X, 1)\right|}{\mathbb{P}(Y = 1 \mid S = 1)}}_{\text{how good is } \hat{\eta}} + 2\underbrace{\frac{\mathbb{E}_{X|S=0}\left|\eta(X, 0) - \hat{\eta}(X, 0)\right|}{\mathbb{P}(Y = 1 \mid S = 0)}}_{\text{how good is } \hat{\eta}}$$

$$+ \underbrace{\frac{\left|(\mathbb{E}_{X|S=1} - \hat{\mathbb{E}}_{X|S=1})\hat{\eta}(X, 1)g(X, 1)\right|}{\mathbb{E}_{X|S=1}\hat{\eta}(X, 1)}}_{\text{empirical process}} + \underbrace{\frac{\left|(\mathbb{E}_{X|S=0} - \hat{\mathbb{E}}_{X|S=0})\hat{\eta}(X, 0)g(X, 0)\right|}{\mathbb{E}_{X|S=0}\hat{\eta}(X, 0)}}_{\text{empirical process}}$$

$$+ \frac{\left|(\hat{\mathbb{E}}_{X|S=1} - \mathbb{E}_{X|S=1})\hat{\eta}(X, 1)\right|}{\mathbb{E}_{X|S=1}\hat{\eta}(X, 1)} + \frac{\left|(\hat{\mathbb{E}}_{X|S=0} - \mathbb{E}_{X|S=0})\hat{\eta}(X, 0)\right|}{\mathbb{E}_{X|S=0}\hat{\eta}(X, 0)} \ .$$

Let us elaborate on the above result. The second and the third terms are responsible for the estimation of $\eta$ and can be controlled in various parametric on nonparametric models. The third and the fourth terms can be handled with the theory of empirical processes in the considered classifier $g$ is data

dependent. The last two terms can be handled conditionally on the first labeled samples by the use of the multiplicative Chernoff's bound or (if we do not mind losing a constant factor of 2) can be handled by the empirical process used to bound the third and the fourth terms.

The next lemma gives an upper bound on the empirical processes of Lemma C.1.

**Lemma C.2.** *There exists a constant $C > 0$ that depends only on $\mathbb{P}(S = 0)$ and $\mathbb{P}(S = 1)$ such that almost surely for all $s \in \{0, 1\}$ we have*

$$\mathbb{E}_{\mathcal{D}_N} \sup_{t \in [0,1]} \left| (\mathbb{E}_{X|S=s} - \hat{\mathbb{E}}_{X|S=s}) \hat{\eta}(X, s) \mathbf{1}_{\{t \leq \hat{\eta}(X,s)\}} \right| \leq C \sqrt{\frac{1}{N}} \ .$$

The next result is obvious, yet, is used several times in our proof, that is why we present it separately.

**Lemma C.3.** *For any functions $h_1, h_0 : \mathbb{R}^d \to [0, 1]$, any $\theta \in \mathbb{R}$, any $a_1, a_0, b_1, b_0 \in (0, 1)$ we have*

$$\mathbb{E}_{X|S=1} \left[ \frac{\theta h_1(X)}{a_1} \mathbf{1}_{\left\{ b_1(2h_1(X)-1) - \frac{\theta h_1(X)}{a_1} \geq 0 \right\}} \right]$$
$$= \mathbb{E}_{X|S=1} \left[ (2h_1(X) - 1) \mathbf{1}_{\left\{ b_1(2h_1(X)-1) - \frac{\theta h_1(X)}{a_1} \geq 0 \right\}} \right] b_1$$
$$- \mathbb{E}_{X|S=1} \left( b_1(2h_1(X) - 1) - \frac{\theta h_1(X)}{a_1} \right)_+ \ ,$$

$$\mathbb{E}_{X|S=0} \left[ \frac{\theta h_0(X)}{a_0} \mathbf{1}_{\left\{ b_0(2h_0(X)-1) + \frac{\theta h_0(X)}{a_0} \geq 0 \right\}} \right]$$
$$= -\mathbb{E}_{X|S=0} \left[ (2h_0(X) - 1) \mathbf{1}_{\left\{ b_0(2h_0(X)-1) + \frac{\theta h_0(X)}{a_0} \geq 0 \right\}} \right] b_0$$
$$+ \mathbb{E}_{X|S=0} \left( b_0(2h_0(X) - 1) + \frac{\theta h_0(X)}{a_0} \right)_+ \ ,$$

*moreover, the expectation $\mathbb{E}_{X|S=s}$ can be replaced by $\hat{\mathbb{E}}_{X|S=s}$ for all $s \in \{0, 1\}$.*

## C.1 Proof of asymptotic fairness (Part I of Theorem 4.5)

*Proof.* The first step is to show that under Assumption 4.3 the term $\hat{\Delta}(\hat{g}, \mathbb{P})$ cannot be too big. Indeed, notice that for every $\theta \in [-2, 2]$, thanks to the triangle inequality we can write almost surely

$$\hat{\Delta}(\hat{g}_\theta, \mathbb{P}) \leq \left| \frac{\mathbb{E}_{X|S=1} \hat{\eta}(X, 1) \hat{g}_\theta(X, 1)}{\mathbb{E}_{X|S=1} \hat{\eta}(X, 1)} - \frac{\mathbb{E}_{X|S=0} \hat{\eta}(X, 0) \hat{g}_\theta(X, 0)}{\mathbb{E}_{X|S=0} \hat{\eta}(X, 0)} \right|$$
$$+ \left| \frac{\mathbb{E}_{X|S=1} \hat{\eta}(X, 1) \hat{g}_\theta(X, 1)}{\mathbb{E}_{X|S=1} \hat{\eta}(X, 1)} - \frac{\hat{\mathbb{E}}_{X|S=1} \hat{\eta}(X, 1) \hat{g}_\theta(X, 1)}{\hat{\mathbb{E}}_{X|S=1} \hat{\eta}(X, 1)} \right| \quad (8)$$
$$+ \left| \frac{\mathbb{E}_{X|S=0} \hat{\eta}(X, 0) \hat{g}_\theta(X, 0)}{\mathbb{E}_{X|S=0} \hat{\eta}(X, 0)} - \frac{\hat{\mathbb{E}}_{X|S=0} \hat{\eta}(X, 0) \hat{g}_\theta(X, 0)}{\hat{\mathbb{E}}_{X|S=0} \hat{\eta}(X, 0)} \right| \ .$$

Our goal is to take care of each of the three terms appearing on the right hand side of the inequality. The technique used for the second and the third term is identical, whereas the first term is a bit more involved. Let us start with the second term on the right hand side of Eq. (8). For this term we can

write almost surely

$$\left| \frac{\mathbb{E}_{X|S=1}\hat{\eta}(X,1)\hat{g}_\theta(X,1)}{\mathbb{E}_{X|S=1}\hat{\eta}(X,1)} - \frac{\hat{\mathbb{E}}_{X|S=1}\hat{\eta}(X,1)\hat{g}_\theta(X,1)}{\hat{\mathbb{E}}_{X|S=1}\hat{\eta}(X,1)} \right|$$

$$\leq \left| \frac{\mathbb{E}_{X|S=1}\hat{\eta}(X,1)\hat{g}_\theta(X,1)}{\mathbb{E}_{X|S=1}\hat{\eta}(X,1)} - \frac{\hat{\mathbb{E}}_{X|S=1}\hat{\eta}(X,1)\hat{g}_\theta(X,1)}{\mathbb{E}_{X|S=1}\hat{\eta}(X,1)} \right|$$

$$+ \left| \frac{\hat{\mathbb{E}}_{X|S=1}\hat{\eta}(X,1)\hat{g}_\theta(X,1)}{\hat{\mathbb{E}}_{X|S=1}\hat{\eta}(X,1)} - \frac{\hat{\mathbb{E}}_{X|S=1}\hat{\eta}(X,1)\hat{g}_\theta(X,1)}{\mathbb{E}_{X|S=1}\hat{\eta}(X,1)} \right|$$

$$= \frac{\left| (\mathbb{E}_{X|S=1} - \hat{\mathbb{E}}_{X|S=1})\hat{\eta}(X,1)\hat{g}_\theta(X,1) \right|}{\mathbb{E}_{X|S=1}\hat{\eta}(X,1)}$$

$$+ \underbrace{\frac{\hat{\mathbb{E}}_{X|S=1}\hat{\eta}(X,1)\hat{g}_\theta(X,1)}{\hat{\mathbb{E}}_{X|S=1}\hat{\eta}(X,1)}}_{\leq 1} \frac{\left| (\mathbb{E}_{X|S=1} - \hat{\mathbb{E}}_{X|S=1})\hat{\eta}(X,1)\mathbf{1}_{\{0 \leq \hat{\eta}(X,1)\}} \right|}{\mathbb{E}_{X|S=1}\hat{\eta}(X,1)}$$

$$\leq 2 \frac{\sup_{t\in[0,1]}\left| (\mathbb{E}_{X|S=1} - \hat{\mathbb{E}}_{X|S=1})\hat{\eta}(X,1)\mathbf{1}_{\{t \leq \hat{\eta}(X,1)\}} \right|}{\mathbb{E}_{X|S=1}\hat{\eta}(X,1)} \;,$$

where the last inequality follows from the fact that $\hat{g}_\theta$ is a thresholding rule. Similarly, we show that the third term in Eq. (8) admits the following bound almost surely

$$\left| \frac{\mathbb{E}_{X|S=0}\hat{\eta}(X,0)\hat{g}_\theta(X,0)}{\mathbb{E}_{X|S=0}\hat{\eta}(X,0)} - \frac{\hat{\mathbb{E}}_{X|S=0}\hat{\eta}(X,0)\hat{g}_\theta(X,0)}{\hat{\mathbb{E}}_{X|S=0}\hat{\eta}(X,0)} \right|$$

$$\leq 2 \frac{\sup_{t\in[0,1]}\left| (\mathbb{E}_{X|S=0} - \hat{\mathbb{E}}_{X|S=0})\hat{\eta}(X,0)\mathbf{1}_{\{t \leq \hat{\eta}(X,0)\}} \right|}{\mathbb{E}_{X|S=0}\hat{\eta}(X,0)} \;.$$

Therefore, we arrive at the following bound on $\hat{\Delta}(\hat{g}_\theta, \mathbb{P})$ which holds almost surely

$$\hat{\Delta}(\hat{g}_\theta, \mathbb{P}) \leq \left| \frac{\mathbb{E}_{X|S=1}\hat{\eta}(X,1)\hat{g}_\theta(X,1)}{\mathbb{E}_{X|S=1}\hat{\eta}(X,1)} - \frac{\mathbb{E}_{X|S=0}\hat{\eta}(X,0)\hat{g}_\theta(X,0)}{\mathbb{E}_{X|S=0}\hat{\eta}(X,0)} \right| \tag{9}$$

$$+ 2 \frac{\sup_{t\in[0,1]}\left| (\mathbb{E}_{X|S=1} - \hat{\mathbb{E}}_{X|S=1})\hat{\eta}(X,1)\mathbf{1}_{\{t \leq \hat{\eta}(X,1)\}} \right|}{\mathbb{E}_{X|S=1}\hat{\eta}(X,1)}$$

$$+ 2 \frac{\sup_{t\in[0,1]}\left| (\mathbb{E}_{X|S=0} - \hat{\mathbb{E}}_{X|S=0})\hat{\eta}(X,0)\mathbf{1}_{\{t \leq \hat{\eta}(X,0)\}} \right|}{\mathbb{E}_{X|S=0}\hat{\eta}(X,0)} \;.$$

This is one of the moments when we make use of Assumption 4.3. Thanks to the continuity we can be sure that for every possible unlabeled sample $\mathcal{D}_N$, there exists $\theta'(\mathcal{D}_N)$ such that

$$\frac{\mathbb{E}_{X|S=1}\hat{\eta}(X,1)\hat{g}_{\theta'(\mathcal{D}_N)}(X,1)}{\mathbb{E}_{X|S=1}\hat{\eta}(X,1)} = \frac{\mathbb{E}_{X|S=0}\hat{\eta}(X,0)\hat{g}_{\theta'(\mathcal{D}_N)}(X,0)}{\mathbb{E}_{X|S=0}\hat{\eta}(X,0)} \;.$$

Indeed, for every possible unlabeled sample $\mathcal{D}_N$ on the left hand side we have a continuous decreasing of $\theta$ function and on the right hand side we have a continuous increasing function of $\theta$. Therefore, such a value $\theta'(\mathcal{D}_N)$ exists.

Taking infimum over $\theta \in [-2, 2]$ on both sides of Equation (9) we obtain

$$\hat{\Delta}(\hat{g}, \mathbb{P}) = \hat{\Delta}(\hat{g}_{\hat{\theta}}, \mathbb{P}) \leq 2 \frac{\sup_{t\in[0,1]}\left| (\mathbb{E}_{X|S=1} - \hat{\mathbb{E}}_{X|S=1})\hat{\eta}(X,1)\mathbf{1}_{\{t \leq \hat{\eta}(X,1)\}} \right|}{\mathbb{E}_{X|S=1}\hat{\eta}(X,1)} \tag{10}$$

$$+ 2 \frac{\sup_{t\in[0,1]}\left| (\mathbb{E}_{X|S=0} - \hat{\mathbb{E}}_{X|S=0})\hat{\eta}(X,0)\mathbf{1}_{\{t \leq \hat{\eta}(X,0)\}} \right|}{\mathbb{E}_{X|S=0}\hat{\eta}(X,0)} \;.$$

Using Lemma C.1 and applying it to $\hat{g}$ we immediately obtain almost surely

$$\Delta(\hat{g}, \mathbb{P}) \leq 4 \frac{\sup_{t \in [0,1]} \left| (\mathbb{E}_{X|S=1} - \hat{\mathbb{E}}_{X|S=1}) \hat{\eta}(X,1) \mathbf{1}_{\{t \leq \hat{\eta}(X,1)\}} \right|}{\mathbb{E}_{X|S=1} \hat{\eta}(X,1)}$$

$$+ 4 \frac{\sup_{t \in [0,1]} \left| (\mathbb{E}_{X|S=0} - \hat{\mathbb{E}}_{X|S=0}) \hat{\eta}(X,0) \mathbf{1}_{\{t \leq \hat{\eta}(X,0)\}} \right|}{\mathbb{E}_{X|S=0} \hat{\eta}(X,0)}$$

$$+ 2 \frac{\mathbb{E}_{X|S=1} |\eta(X,1) - \hat{\eta}(X,1)|}{\mathbb{P}(Y=1 \,|\, S=1)} + 2 \frac{\mathbb{E}_{X|S=0} |\eta(X,0) - \hat{\eta}(X,0)|}{\mathbb{P}(Y=1 \,|\, S=0)} \ .$$

Clearly, if $\hat{\eta}$ is a consistent estimator of $\eta$ then the last two terms on the right hand side are converging to zero in expectation as $n \to \infty$. Therefore, it remain to provide an upper bound for the two empirical processes. Recall, that our goal is to obtain consistency in expectation, thus we take expectation *w.r.t.* $\mathcal{D}_n, \mathcal{D}_N$ from both sides of the inequality. Thanks to Lemma C.2 we have for each $s \in \{0,1\}$

$$\mathbb{E}_{\mathcal{D}_N} \sup_{t \in [0,1]} \left| (\mathbb{E}_{X|S=s} - \hat{\mathbb{E}}_{X|S=s}) \hat{\eta}(X,s) \mathbf{1}_{\{t \leq \hat{\eta}(X,s)\}} \right| \leq C \sqrt{\frac{1}{N}} \ .$$

The arguments above imply that there exists an absolute constant $C > 0$ such that

$$\mathbb{E}_{(\mathcal{D}_n, \mathcal{D}_N)}[\Delta(\hat{g}, \mathbb{P})] \leq 2 \frac{\mathbb{E}_{\mathcal{D}_n} \mathbb{E}_{X|S=1} |\eta(X,1) - \hat{\eta}(X,1)|}{\mathbb{P}(Y=1 \,|\, S=1)} + 2 \frac{\mathbb{E}_{\mathcal{D}_n} \mathbb{E}_{X|S=0} |\eta(X,0) - \hat{\eta}(X,0)|}{\mathbb{P}(Y=1 \,|\, S=0)}$$

$$+ C \sqrt{\frac{1}{N}} \mathbb{E}_{\mathcal{D}_n} \frac{1}{\min\{\mathbb{E}_{X|S=1} \hat{\eta}(X,1), \mathbb{E}_{X|S=0} \hat{\eta}(X,0)\}} \ .$$

Using the second item of Assumption 4.1, which states that $\min\{\mathbb{E}_{X|S=1}\hat{\eta}(X,1), \mathbb{E}_{X|S=0}\hat{\eta}(X,0)\} \geq c_{n,N}$ almost surely we conclude. $\qquad\square$

## C.2 Proof of asymptotic optimality (Part II of Theorem 4.5)

In order to show that the risk of the proposed algorithm converges to the risk of the optimal classifier, we follow the strategy of [11], that is, we first introduce an intermediate pseudo-estimator $\tilde{g}$ as follows

$$\tilde{g}(x,1) = \mathbf{1}_{\left\{ \mathbb{P}(S=1) \leq \hat{\eta}(x,1) \left( 2\mathbb{P}(S=1) - \frac{\tilde{\theta}}{\mathbb{E}_{X|S=1}[\hat{\eta}(X,1)]} \right) \right\}} \ , \tag{11}$$

$$\tilde{g}(x,0) = \mathbf{1}_{\left\{ \mathbb{P}(S=0) \leq \hat{\eta}(x,0) \left( 2\mathbb{P}(S=0) + \frac{\tilde{\theta}}{\mathbb{E}_{X|S=0}[\hat{\eta}(X,0)]} \right) \right\}} \ , \tag{12}$$

where $\tilde{\theta}$ is a solution of

$$\frac{\mathbb{E}_{X|S=1} [\hat{\eta}(X,1) \tilde{g}_\theta(X,1)]}{\mathbb{E}_{X|S=1}[\hat{\eta}(X,1)]} = \frac{\mathbb{E}_{X|S=0} [\hat{\eta}(X,0) \tilde{g}_\theta(X,0)]}{\mathbb{E}_{X|S=0}[\hat{\eta}(X,0)]} \ , \tag{13}$$

with $\tilde{g}_\theta$ being defined as for all $x \in \mathbb{R}^d$ as

$$\tilde{g}_\theta(x,1) = \mathbf{1}_{\left\{ 1 \leq \hat{\eta}(X,1) \left( 2 - \frac{\theta}{\mathbb{E}_{X|S=1}[\hat{\eta}(X,1)]\mathbb{P}(S=1)} \right) \right\}} \ ,$$

$$\tilde{g}_\theta(x,0) = \mathbf{1}_{\left\{ 1 \leq \hat{\eta}(X,0) \left( 2 + \frac{\theta}{\mathbb{E}_{X|S=0}[\hat{\eta}(X,0)]\mathbb{P}(S=0)} \right) \right\}} \ .$$

Note that thanks to Assumption 4.3 such a value $\tilde{\theta}$ always exists.

Intuitively, the classifier $\tilde{g}$ knows the marginal distribution of $(X, S)$, that is, it knows both $\mathbb{P}_{X|s}$ and $\mathbb{P}_S$. It is seen as an idealized version of $\hat{g}$, where the uncertainty is only induced by the lack of knowledge of the regression function $\eta$. We upper bound the excess risk in two steps. In the first step we upper bound $\mathcal{R}(\tilde{g}) - \mathcal{R}(g^*)$ and on the second we upper bound the difference $\mathcal{R}(\hat{g}) - \mathcal{R}(\tilde{g})$.

**Theorem C.4** (Bound on the pseudo oracle). *Let $\tilde{g}$ be the pseudo oracle classifier defined in Eq. 11 with $\hat{\eta}$ satisfying Assumptions 4.1 and 4.3, then*

$$\lim_{n \to \infty} \mathbb{E}_{\mathcal{D}_n}[\mathcal{R}(\tilde{g})] - \mathcal{R}(g^*) \leq 0 \ .$$

*Proof of Theorem C.4.* First of all, let us rewrite the equation for $\theta^*$ in the following form

$$\mathbb{E}_{X|S=1}\left[\frac{\theta^*\eta(X,1)}{\mathbb{E}_{X|S=1}[\eta(X,1)]}\mathbf{1}_{\left\{\mathbb{P}(S=1)(2\eta(X,1)-1)-\frac{\theta^*\eta(X,1)}{\mathbb{E}_{X|S=1}[\eta(X,1)]}\geq 0\right\}}\right]$$

$$=\mathbb{E}_{X|S=0}\left[\frac{\theta^*\eta(X,0)}{\mathbb{E}_{X|S=0}[\eta(X,0)]}\mathbf{1}_{\left\{\mathbb{P}(S=0)(2\eta(X,0)-1)+\frac{\theta^*\eta(X,0)}{\mathbb{E}_{X|S=0}[\eta(X,0)]}\geq 0\right\}}\right] \ .$$

Using Lemma C.3 with $h_s(\cdot)\equiv\eta(\cdot,s), a_s=\mathbb{E}_{X|S=1}[h_s(\cdot)], b_s=\mathbb{P}(S=s)$ for $s\in\{0,1\}$ we get

$$\mathbb{P}(S=1)\mathbb{E}_{X|S=1}[(2\eta(X,1)-1)g^*(X,1)]$$

$$-\mathbb{E}_{X|S=1}\left(\mathbb{P}(S=1)(2\eta(X,1)-1)-\frac{\theta^*\eta(X,1)}{\mathbb{E}_{X|S=1}[\eta(X,1)]}\right)_+$$

$$=-\mathbb{P}(S=0)\mathbb{E}_{X|S=0}[(2\eta(X,0)-1)g^*(X,0)]$$

$$+\mathbb{E}_{X|S=0}\left(\mathbb{P}(S=0)(2\eta(X,0)-1)+\frac{\theta^*\eta(X,0)}{\mathbb{E}_{X|S=0}[\eta(X,0)]}\right)_+ \ .$$

Rearranging the terms we can arrive at

$$\mathbb{P}(S=1)\mathbb{E}_{X|S=1}[(2\eta(X,1)-1)g^*(X,1)]+\mathbb{P}(S=0)\mathbb{E}_{X|S=0}[(2\eta(X,0)-1)g^*(X,0)]$$

$$=\mathbb{E}_{X|S=1}\left(\mathbb{P}(S=1)(2\eta(X,1)-1)-\frac{\theta^*\eta(X,1)}{\mathbb{E}_{X|S=1}[\eta(X,1)]}\right)_+$$

$$+\mathbb{E}_{X|S=0}\left(\mathbb{P}(S=0)(2\eta(X,0)-1)+\frac{\theta^*\eta(X,0)}{\mathbb{E}_{X|S=0}[\eta(X,0)]}\right)_+ \ .$$

Notice that the left hand side of the above equality can be written as

$$\mathbb{E}_{(X,S)}[(2\eta(X,S)-1)g^*(X,S)]$$

$$=\mathbb{E}_{X|S=1}\left(\mathbb{P}(S=1)(2\eta(X,1)-1)-\frac{\theta^*\eta(X,1)}{\mathbb{E}_{X|S=1}[\eta(X,1)]}\right)_+ \tag{14}$$

$$+\mathbb{E}_{X|S=0}\left(\mathbb{P}(S=0)(2\eta(X,0)-1)+\frac{\theta^*\eta(X,0)}{\mathbb{E}_{X|S=0}[\eta(X,0)]}\right)_+ \ .$$

Thus, combining the previous equality with the expression of the risk from Lemma B.3 we get

$$\mathcal{R}(g^*)=\mathbb{E}_{(X,S)}[\eta(X,S)]-\mathbb{E}_{X|S=1}\left(\mathbb{P}(S=1)(2\eta(X,1)-1)-\frac{\theta^*\eta(X,1)}{\mathbb{E}_{X|S=1}[\eta(X,1)]}\right)_+ \tag{15}$$

$$-\mathbb{E}_{X|S=0}\left(\mathbb{P}(S=0)(2\eta(X,0)-1)+\frac{\theta^*\eta(X,0)}{\mathbb{E}_{X|S=0}[\eta(X,0)]}\right)_+ \ .$$

Step-wise similar argument yields that for the pseudo-oracle $\tilde{g}$ we can write

$$\mathbb{E}_{(X,S)}[(2\hat{\eta}(X,S)-1)\tilde{g}(X,S)]$$

$$=\mathbb{E}_{X|S=1}\left(\mathbb{P}(S=1)(2\hat{\eta}(X,1)-1)-\frac{\tilde{\theta}\hat{\eta}(X,1)}{\mathbb{E}_{X|S=1}[\hat{\eta}(X,1)]}\right)_+ \tag{16}$$

$$+\mathbb{E}_{X|S=0}\left(\mathbb{P}(S=0)(2\hat{\eta}(X,0)-1)+\frac{\tilde{\theta}\hat{\eta}(X,0)}{\mathbb{E}_{X|S=0}[\hat{\eta}(X,0)]}\right)_+ \ .$$

Moreover, its risk satisfies

$$\mathcal{R}(\tilde{g})=\mathbb{E}_{(X,S)}[\eta(X,S)]-\mathbb{E}_{(X,S)}[(2\eta(X,S)-1)\tilde{g}(X,S)] \tag{17}$$

$$\leq \mathbb{E}_{(X,S)}[\eta(X,S)]-\mathbb{E}_{(X,S)}[(2\hat{\eta}(X,S)-1)\tilde{g}(X,S)]+2\mathbb{E}_{(X,S)}|\hat{\eta}(X,S)-\eta(X,S)| \ .$$

Therefore, combining Eq. (15) with Eq. (17), we can write for the excess risk

$$\mathcal{R}(\tilde{g}) - \mathcal{R}(g^*) \leq 2\mathbb{E}_{(X,S)}\left|\hat{\eta}(X,S) - \eta(X,S)\right|$$

$$+ \mathbb{E}_{X|S=1}\left(\mathbb{P}(S=1)(2\eta(X,1)-1) - \frac{\theta^*\eta(X,1)}{\mathbb{E}_{X|S=1}[\eta(X,1)]}\right)_+$$

$$- \mathbb{E}_{X|S=1}\left(\mathbb{P}(S=1)(2\hat{\eta}(X,1)-1) - \frac{\tilde{\theta}\hat{\eta}(X,1)}{\mathbb{E}_{X|S=1}[\hat{\eta}(X,1)]}\right)_+$$

$$+ \mathbb{E}_{X|S=0}\left(\mathbb{P}(S=0)(2\eta(X,0)-1) + \frac{\theta^*\eta(X,0)}{\mathbb{E}_{X|S=0}[\eta(X,0)]}\right)_+$$

$$- \mathbb{E}_{X|S=0}\left(\mathbb{P}(S=0)(2\hat{\eta}(X,0)-1) + \frac{\tilde{\theta}\hat{\eta}(X,0)}{\mathbb{E}_{X|S=0}[\hat{\eta}(X,0)]}\right)_+ \quad .$$

Recall that $\theta^*$ is a minimizer of

$$\mathbb{E}_{X|S=1}\left(\mathbb{P}(S=1)(2\eta(X,1)-1) - \frac{\theta\eta(X,1)}{\mathbb{E}_{X|S=1}[\eta(X,1)]}\right)_+$$

$$+ \mathbb{E}_{X|S=0}\left(\mathbb{P}(S=0)(2\eta(X,0)-1) + \frac{\theta\eta(X,0)}{\mathbb{E}_{X|S=0}[\eta(X,0)]}\right)_+ \quad ,$$

thus we can replace $\theta^*$ by $\tilde{\theta}$ and obtain the following upper bound

$$\mathcal{R}(\tilde{g}) - \mathcal{R}(g^*) \leq 2\mathbb{E}_{(X,S)}\left|\hat{\eta}(X,S) - \eta(X,S)\right|$$

$$+ \mathbb{E}_{X|S=1}\left(\mathbb{P}(S=1)(2\eta(X,1)-1) - \frac{\tilde{\theta}\eta(X,1)}{\mathbb{E}_{X|S=1}[\eta(X,1)]}\right)_+$$

$$- \mathbb{E}_{X|S=1}\left(\mathbb{P}(S=1)(2\hat{\eta}(X,1)-1) - \frac{\tilde{\theta}\hat{\eta}(X,1)}{\mathbb{E}_{X|S=1}[\hat{\eta}(X,1)]}\right)_+$$

$$+ \mathbb{E}_{X|S=0}\left(\mathbb{P}(S=0)(2\eta(X,0)-1) + \frac{\tilde{\theta}\eta(X,0)}{\mathbb{E}_{X|S=0}[\eta(X,0)]}\right)_+$$

$$- \mathbb{E}_{X|S=0}\left(\mathbb{P}(S=0)(2\hat{\eta}(X,0)-1) + \frac{\tilde{\theta}\hat{\eta}(X,0)}{\mathbb{E}_{X|S=0}[\hat{\eta}(X,0)]}\right)_+ \quad .$$

Since, for all $x, y \in \mathbb{R}$ we have $(x)_+ - (y)_+ \leq (x-y)_+ \leq |x-y|$ we get

$$\mathcal{R}(\tilde{g}) - \mathcal{R}(g^*) \leq 4\mathbb{E}_{(X,S)}\left|\hat{\eta}(X,S) - \eta(X,S)\right|$$

$$+ \mathbb{E}_{X|S=1}|\tilde{\theta}|\left|\frac{\hat{\eta}(X,1)}{\mathbb{E}_{X|S=1}[\hat{\eta}(X,1)]} - \frac{\eta(X,1)}{\mathbb{E}_{X|S=1}[\eta(X,1)]}\right|$$

$$+ \mathbb{E}_{X|S=0}|\tilde{\theta}|\left|\frac{\hat{\eta}(X,0)}{\mathbb{E}_{X|S=0}[\hat{\eta}(X,0)]} - \frac{\eta(X,0)}{\mathbb{E}_{X|S=0}[\eta(X,0)]}\right| \quad .$$

For the same reason why $|\theta^*| \leq 2$ we have $|\tilde{\theta}| \leq 2$, thus for all $s \in \{0,1\}$ we have

$$\mathbb{E}_{X|S=s}|\tilde{\theta}|\left|\frac{\hat{\eta}(X,s)}{\mathbb{E}_{X|S=s}[\hat{\eta}(X,s)]} - \frac{\eta(X,s)}{\mathbb{E}_{X|S=s}[\eta(X,s)]}\right|$$

$$\leq 2\mathbb{E}_{X|S=s}\left|\frac{\hat{\eta}(X,s)}{\mathbb{E}_{X|S=s}[\hat{\eta}(X,s)]} - \frac{\eta(X,s)}{\mathbb{E}_{X|S=s}[\eta(X,s)]}\right|$$

$$\leq 2\mathbb{E}_{X|S=s}\left|\frac{\hat{\eta}(X,s)}{\mathbb{E}_{X|S=s}[\eta(X,s)]} - \frac{\eta(X,s)}{\mathbb{E}_{X|S=s}[\eta(X,s)]}\right|$$

$$+ 2\mathbb{E}_{X|S=s}\left|\frac{\hat{\eta}(X,s)}{\mathbb{E}_{X|S=s}[\hat{\eta}(X,s)]} - \frac{\hat{\eta}(X,s)}{\mathbb{E}_{X|S=s}[\eta(X,s)]}\right|$$

$$\leq 4\frac{\mathbb{E}_{X|S=s}\left|\eta(X,s) - \hat{\eta}(X,s)\right|}{\mathbb{E}_{X|S=s}[\eta(X,s)]} \quad .$$

Thanks to Assumption 4.1, these terms converge to zero in expectation. $\qquad\square$

**Theorem C.5.** *Let $\hat{g}$ be the proposed classifier with $\hat{\eta}$ satisfying Assumptions 4.1 and 4.3, then*

$$\lim_{n\to\infty}\mathbb{E}_{(\mathcal{D}_n,\mathcal{D}_N)}[\mathcal{R}(\hat{g})-\mathcal{R}(\tilde{g})]\le 0 \ .$$

*Proof.* Our goal is to upper bound the quantity $\mathbb{E}_{(\mathcal{D}_n,\mathcal{D}_N)}\mathcal{R}(\hat{g})-\mathcal{R}(\tilde{g})$. We start by providing a bound on $\mathcal{R}(\hat{g})-\mathcal{R}(\tilde{g})$ which holds almost surely. Recall the equality of Equation (16)

$$\mathbb{E}_{(X,S)}[(2\hat{\eta}(X,S)-1)\tilde{g}(X,S)]$$
$$= \mathbb{E}_{X|S=1}\left(\mathbb{P}(S=1)(2\hat{\eta}(X,1)-1)-\frac{\tilde{\theta}\hat{\eta}(X,1)}{\mathbb{E}_{X|S=1}[\hat{\eta}(X,1)]}\right)_+$$
$$+ \mathbb{E}_{X|S=0}\left(\mathbb{P}(S=0)(2\hat{\eta}(X,0)-1)+\frac{\tilde{\theta}\hat{\eta}(X,0)}{\mathbb{E}_{X|S=0}[\hat{\eta}(X,0)]}\right)_+ \ .$$

Using this and the expression of the risk given in Lemma B.3 we can obtain the following lower bound on the risk of $\tilde{g}$

$$\begin{aligned}
\mathcal{R}(\tilde{g}) &= \mathbb{E}_{(X,S)}[\eta(X,S)]-\mathbb{E}_{(X,S)}[(2\eta(X,S)-1)\tilde{g}(X,S)]\\
&\ge \mathbb{E}_{(X,S)}[\eta(X,S)]-\mathbb{E}_{(X,S)}[(2\hat{\eta}(X,S)-1)\tilde{g}(X,S)]-2\mathbb{E}_{(X,S)}|\hat{\eta}(X,S)-\eta(X,S)|\\
&= \mathbb{E}_{(X,S)}[\eta(X,S)]-2\mathbb{E}_{(X,S)}|\hat{\eta}(X,S)-\eta(X,S)| \qquad\qquad (18)\\
&\quad -\mathbb{E}_{X|S=1}\left(\mathbb{P}(S=1)(2\hat{\eta}(X,1)-1)-\frac{\tilde{\theta}\hat{\eta}(X,1)}{\mathbb{E}_{X|S=1}[\hat{\eta}(X,1)]}\right)_+\\
&\quad -\mathbb{E}_{X|S=0}\left(\mathbb{P}(S=0)(2\hat{\eta}(X,0)-1)+\frac{\tilde{\theta}\hat{\eta}(X,0)}{\mathbb{E}_{X|S=0}[\hat{\eta}(X,0)]}\right)_+ \ .
\end{aligned}$$

We have thanks to Lemma C.3 used with $h_s(\cdot)=\hat{\eta}(\cdot,s), a_s=\hat{\mathbb{E}}_{X|S=s}[h_s(X)], b_s=\hat{\mathbb{P}}(S=s)$ for all $s\in\{0,1\}$

$$\frac{\hat{\mathbb{E}}_{X|S=1}\hat{\theta}\hat{\eta}(X,1)\hat{g}(X,1)}{\hat{\mathbb{E}}_{X|S=1}\hat{\eta}(X,1)}=\hat{\mathbb{E}}_{X|S=1}[(2\hat{\eta}(X,1)-1)\hat{g}(X,1)]\hat{\mathbb{P}}(S=1) \qquad (19)$$
$$-\hat{\mathbb{E}}_{X|S=1}\left(\hat{\mathbb{P}}(S=1)(2\hat{\eta}(X,1)-1)-\frac{\hat{\theta}\hat{\eta}(X,1)}{\hat{\mathbb{E}}_{X|S=1}[\hat{\eta}(X,1)]}\right)_+ \ ,$$

and

$$\frac{\hat{\mathbb{E}}_{X|S=0}\hat{\theta}\hat{\eta}(X,0)\hat{g}(X,0)}{\hat{\mathbb{E}}_{X|S=0}\hat{\eta}(X,0)}=-\hat{\mathbb{E}}_{X|S=0}[(2\hat{\eta}(X,0)-1)\hat{g}(X,0)]\hat{\mathbb{P}}(S=0) \qquad (20)$$
$$+\hat{\mathbb{E}}_{X|S=0}\left(\hat{\mathbb{P}}(S=0)(2\hat{\eta}(X,0)-1)+\frac{\hat{\theta}\hat{\eta}(X,0)}{\hat{\mathbb{E}}_{X|S=0}[\hat{\eta}(X,0)]}\right)_+ \ .$$

Recall, that thanks to Definition 3.2 of the empirical unfairness we have

$$|\hat{\theta}|\hat{\Delta}(\hat{g},\mathbb{P})=\left|\frac{\hat{\mathbb{E}}_{X|S=0}\hat{\theta}\hat{\eta}(X,0)\hat{g}(X,0)}{\hat{\mathbb{E}}_{X|S=0}\hat{\eta}(X,0)}-\frac{\hat{\mathbb{E}}_{X|S=1}\hat{\theta}\hat{\eta}(X,1)\hat{g}(X,1)}{\hat{\mathbb{E}}_{X|S=1}\hat{\eta}(X,1)}\right| \ .$$

Since, $|\hat{\theta}| \leq 2$, subtracting Eq. (20) from Eq. (19) and taking absolute value combined with the triangle inequality we get

$$
\begin{aligned}
&\hat{\mathbb{E}}_{(X,S)}(2\hat{\eta}(X,S)-1)\hat{g}(X,S) \\
&= \hat{\mathbb{E}}_{X|S=0}[(2\hat{\eta}(X,0)-1)\hat{g}(X,0)]\hat{\mathbb{P}}(S=0) + \hat{\mathbb{E}}_{X|S=1}[(2\hat{\eta}(x,1)-1)\hat{g}(X,1)]\hat{\mathbb{P}}(S=1) \quad (21) \\
&\geq -2\hat{\Delta}(\hat{g},\mathbb{P}) + \hat{\mathbb{E}}_{X|S=0}\left( \hat{\mathbb{P}}(S=0)(2\hat{\eta}(X,0)-1) + \frac{\hat{\theta}\hat{\eta}(X,0)}{\hat{\mathbb{E}}_{X|S=0}[\hat{\eta}(X,0)]} \right)_{+} \\
&\quad + \hat{\mathbb{E}}_{X|S=1}\left( \hat{\mathbb{P}}(S=1)(2\hat{\eta}(X,1)-1) - \frac{\hat{\theta}\hat{\eta}(X,1)}{\hat{\mathbb{E}}_{X|S=1}[\hat{\eta}(X,1)]} \right)_{+} .
\end{aligned}
$$

Note that using the bound above we can get the following upper bound on the risk of the proposed classifier

$$
\begin{aligned}
\mathcal{R}(\hat{g}) &= \mathbb{E}_{(X,S)}[\eta(X,S)] - \mathbb{E}_{(X,S)}[(2\eta(X,S)-1)\hat{g}(X,S)] \\
&\leq \mathbb{E}_{(X,S)}[\eta(X,S)] - \mathbb{E}_{(X,S)}[(2\hat{\eta}(X,S)-1)\hat{g}(X,S)] \\
&\quad + 2\mathbb{E}_{(X,S)}|\eta(X,S)-\hat{\eta}(X,S)| \qquad \text{(replaced } \eta \text{ by } \hat{\eta}) \\
&\leq \mathbb{E}_{(X,S)}[\eta(X,S)] - \hat{\mathbb{E}}_{(X,S)}[(2\hat{\eta}(X,S)-1)\hat{g}(X,S)] + 2\mathbb{E}_{(X,S)}|\eta(X,S)-\hat{\eta}(X,S)| \\
&\quad + \left| (\mathbb{E}_{(X,S)} - \hat{\mathbb{E}}_{(X,S)})[(2\hat{\eta}(X,S)-1)\hat{g}(X,S)] \right| \qquad \text{(replaced } \mathbb{E}_{(X,S)} \text{ by } \hat{\mathbb{E}}_{(X,S)}) \\
&\leq \mathbb{E}_{(X,S)}[\eta(X,S)] - \hat{\mathbb{E}}_{(X,S)}[(2\hat{\eta}(X,S)-1)\hat{g}(X,S)] + 2\mathbb{E}_{(X,S)}|\eta(X,S)-\hat{\eta}(X,S)| \\
&\quad + \sup_{t\in[0,1]} \left| (\mathbb{E}_{(X,S)} - \hat{\mathbb{E}}_{(X,S)})[(2\hat{\eta}(X,S)-1)\mathbf{1}_{\{t\leq\hat{\eta}(X,S)\}}] \right| \qquad \text{(since } \hat{g} \text{ is thresholding)} \\
&\leq \mathbb{E}_{(X,S)}[\eta(X,S)] + 2\mathbb{E}_{(X,S)}|\eta(X,S)-\hat{\eta}(X,S)| \\
&\quad + 2\hat{\Delta}(\hat{g},\mathbb{P}) + \sup_{t\in[0,1]} \left| (\mathbb{E}_{(X,S)} - \hat{\mathbb{E}}_{(X,S)})[(2\hat{\eta}(X,S)-1)\mathbf{1}_{\{t\leq\hat{\eta}(X,S)\}}] \right| \\
&\quad - \hat{\mathbb{E}}_{X|S=0}\left( \hat{\mathbb{P}}(S=0)(2\hat{\eta}(X,0)-1) + \frac{\hat{\theta}\hat{\eta}(X,0)}{\hat{\mathbb{E}}_{X|S=0}[\hat{\eta}(X,0)]} \right)_{+} \\
&\quad - \hat{\mathbb{E}}_{X|S=1}\left( \hat{\mathbb{P}}(S=1)(2\hat{\eta}(X,1)-1) - \frac{\hat{\theta}\hat{\eta}(X,1)}{\hat{\mathbb{E}}_{X|S=1}[\hat{\eta}(X,1)]} \right)_{+} \qquad \text{(after Eq. (21))} .
\end{aligned}
$$

Thus, combining this upper bound on $\mathcal{R}(\hat{g})$ with the lower bound on $\mathcal{R}(\tilde{g})$ given in Eq. (18) we arrive at

$$
\begin{aligned}
\mathcal{R}(\hat{g}) - \mathcal{R}(\tilde{g}) &\leq 4\mathbb{E}_{(X,S)}|\eta(X,S)-\hat{\eta}(X,S)| + 2\hat{\Delta}(\hat{g},\mathbb{P}) \\
&\quad + \sup_{t\in[0,1]} \left| (\mathbb{E}_{(X,S)} - \hat{\mathbb{E}}_{(X,S)})[(2\hat{\eta}(X,S)-1)\mathbf{1}_{\{t\leq\hat{\eta}(X,S)\}}] \right| \\
&\quad + \mathbb{E}_{X|S=1}\left( \mathbb{P}(S=1)(2\hat{\eta}(X,1)-1) - \frac{\tilde{\theta}\hat{\eta}(X,1)}{\mathbb{E}_{X|S=1}[\hat{\eta}(X,1)]} \right)_{+} \\
&\quad - \hat{\mathbb{E}}_{X|S=1}\left( \hat{\mathbb{P}}(S=1)(2\hat{\eta}(X,1)-1) - \frac{\hat{\theta}\hat{\eta}(X,1)}{\hat{\mathbb{E}}_{X|S=1}[\hat{\eta}(X,1)]} \right)_{+} \\
&\quad + \mathbb{E}_{X|S=0}\left( \mathbb{P}(S=0)(2\hat{\eta}(X,0)-1) + \frac{\tilde{\theta}\hat{\eta}(X,0)}{\mathbb{E}_{X|S=0}[\hat{\eta}(X,0)]} \right)_{+} \\
&\quad - \hat{\mathbb{E}}_{X|S=0}\left( \hat{\mathbb{P}}(S=0)(2\hat{\eta}(X,0)-1) + \frac{\hat{\theta}\hat{\eta}(X,0)}{\hat{\mathbb{E}}_{X|S=0}[\hat{\eta}(X,0)]} \right)_{+} .
\end{aligned}
$$

Thanks to Lemma C.2 the term $\sup_{t\in[0,1]}\left|(\mathbb{E}_{(X,S)}-\hat{\mathbb{E}}_{(X,S)})[(2\hat{\eta}(X,S)-1)\mathbf{1}_{\{t\le\hat{\eta}(X,S)\}}]\right|$ converges to zero in expectation[6]. Equation (10) with Lemma C.2 gives the convergence to zero of $\hat{\Delta}(\hat{g},\mathbb{P})$ in expectation. Assumption 4.1 tells us that the term $\mathbb{E}_{(X,S)}|\eta(X,S)-\hat{\eta}(X,S)|$ goes to zero in expectation. Thus it only remains to bound the term

$$(*) =\mathbb{E}_{X|S=1}\left(\mathbb{P}(S=1)(2\hat{\eta}(X,1)-1)-\frac{\tilde{\theta}\hat{\eta}(X,1)}{\mathbb{E}_{X|S=1}[\hat{\eta}(X,1)]}\right)_{+}$$

$$-\hat{\mathbb{E}}_{X|S=1}\left(\hat{\mathbb{P}}(S=1)(2\hat{\eta}(X,1)-1)-\frac{\tilde{\theta}\hat{\eta}(X,1)}{\hat{\mathbb{E}}_{X|S=1}[\hat{\eta}(X,1)]}\right)_{+}$$

$$+\mathbb{E}_{X|S=0}\left(\mathbb{P}(S=0)(2\hat{\eta}(X,0)-1)+\frac{\tilde{\theta}\hat{\eta}(X,0)}{\mathbb{E}_{X|S=0}[\hat{\eta}(X,0)]}\right)_{+}$$

$$-\hat{\mathbb{E}}_{X|S=0}\left(\hat{\mathbb{P}}(S=0)(2\hat{\eta}(X,0)-1)+\frac{\tilde{\theta}\hat{\eta}(X,0)}{\hat{\mathbb{E}}_{X|S=0}[\hat{\eta}(X,0)]}\right)_{+}.$$

Notice that (similarly to the case of $\theta^*$) the condition in Eq. (13) on $\tilde{\theta}$ is the first order optimality condition for the minimum of the following function

$$\mathbb{E}_{X|S=1}\left(\mathbb{P}(S=1)(2\hat{\eta}(X,1)-1)-\frac{\theta\hat{\eta}(X,1)}{\mathbb{E}_{X|S=1}[\hat{\eta}(X,1)]}\right)_{+}$$

$$+\mathbb{E}_{X|S=0}\left(\mathbb{P}(S=0)(2\hat{\eta}(X,0)-1)+\frac{\theta\hat{\eta}(X,0)}{\mathbb{E}_{X|S=0}[\hat{\eta}(X,0)]}\right)_{+},$$

thus, the objective evaluated at minimum, that is, at $\tilde{\theta}$ is less or equal than the one evaluated at $\hat{\theta}$. Which implies that in order to upper bound $(*)$ it is sufficient to provide an upper bound on

$$(**) =\mathbb{E}_{X|S=1}\left(\mathbb{P}(S=1)(2\hat{\eta}(X,1)-1)-\frac{\hat{\theta}\hat{\eta}(X,1)}{\mathbb{E}_{X|S=1}[\hat{\eta}(X,1)]}\right)_{+}$$

$$-\hat{\mathbb{E}}_{X|S=1}\left(\hat{\mathbb{P}}(S=1)(2\hat{\eta}(X,1)-1)-\frac{\hat{\theta}\hat{\eta}(X,1)}{\hat{\mathbb{E}}_{X|S=1}[\hat{\eta}(X,1)]}\right)_{+}$$

$$+\mathbb{E}_{X|S=0}\left(\mathbb{P}(S=0)(2\hat{\eta}(X,0)-1)+\frac{\hat{\theta}\hat{\eta}(X,0)}{\mathbb{E}_{X|S=0}[\hat{\eta}(X,0)]}\right)_{+}$$

$$-\hat{\mathbb{E}}_{X|S=0}\left(\hat{\mathbb{P}}(S=0)(2\hat{\eta}(X,0)-1)+\frac{\hat{\theta}\hat{\eta}(X,0)}{\hat{\mathbb{E}}_{X|S=0}[\hat{\eta}(X,0)]}\right)_{+},$$

where we replaced $\tilde{\theta}$ by $\hat{\theta}$ thanks to the optimality of $\tilde{\theta}$. Let us define

$$(\triangle) =\mathbb{E}_{X|S=1}\left(\mathbb{P}(S=1)(2\hat{\eta}(X,1)-1)-\frac{\hat{\theta}\hat{\eta}(X,1)}{\mathbb{E}_{X|S=1}[\hat{\eta}(X,1)]}\right)_{+}$$

$$-\hat{\mathbb{E}}_{X|S=1}\left(\hat{\mathbb{P}}(S=1)(2\hat{\eta}(X,1)-1)-\frac{\hat{\theta}\hat{\eta}(X,1)}{\hat{\mathbb{E}}_{X|S=1}[\hat{\eta}(X,1)]}\right)_{+},$$

$$(\triangle\triangle) =\mathbb{E}_{X|S=0}\left(\mathbb{P}(S=0)(2\hat{\eta}(X,0)-1)+\frac{\hat{\theta}\hat{\eta}(X,0)}{\mathbb{E}_{X|S=0}[\hat{\eta}(X,0)]}\right)_{+}$$

$$-\hat{\mathbb{E}}_{X|S=0}\left(\hat{\mathbb{P}}(S=0)(2\hat{\eta}(X,0)-1)+\frac{\hat{\theta}\hat{\eta}(X,0)}{\hat{\mathbb{E}}_{X|S=0}[\hat{\eta}(X,0)]}\right)_{+}.$$

Both bounds are following similar arguments, we demonstrate it for $(\triangle)$, clearly we have

$$(\triangle) \leq \hat{\mathbb{E}}_{X|S=1}\left(\mathbb{P}(S=1)(2\hat{\eta}(X,1)-1) - \frac{\hat{\theta}\hat{\eta}(X,1)}{\mathbb{E}_{X|S=1}[\hat{\eta}(X,1)]}\right)_+$$

$$- \hat{\mathbb{E}}_{X|S=1}\left(\hat{\mathbb{P}}(S=1)(2\hat{\eta}(X,1)-1) - \frac{\hat{\theta}\hat{\eta}(X,1)}{\hat{\mathbb{E}}_{X|S=1}[\hat{\eta}(X,1)]}\right)_+$$

$$+ \left|(\mathbb{E}_{X|S=1} - \hat{\mathbb{E}}_{X|S=1})\left(\mathbb{P}(S=1)(2\hat{\eta}(X,1)-1) - \frac{\hat{\theta}\hat{\eta}(X,1)}{\mathbb{E}_{X|S=1}[\hat{\eta}(X,1)]}\right)_+\right| \quad .$$

For the first difference on the right hand side of this inequality we can write using the fact that $(x)_+ - (y)_+ \leq |x-y|$ for all $x,y \in \mathbb{R}$ and $|2\hat{\eta}(X,1)-1| \leq 1$ almost surely

$$\hat{\mathbb{E}}_{X|S=1}\left(\mathbb{P}(S=1)(2\hat{\eta}(X,1)-1) - \frac{\hat{\theta}\hat{\eta}(X,1)}{\mathbb{E}_{X|S=1}[\hat{\eta}(X,1)]}\right)_+$$

$$- \hat{\mathbb{E}}_{X|S=1}\left(\hat{\mathbb{P}}(S=1)(2\hat{\eta}(X,1)-1) - \frac{\hat{\theta}\hat{\eta}(X,1)}{\hat{\mathbb{E}}_{X|S=1}[\hat{\eta}(X,1)]}\right)_+$$

$$\leq \left|\mathbb{P}(S=1) - \hat{\mathbb{P}}(S=1)\right| + |\hat{\theta}| \left|\frac{\hat{\mathbb{E}}_{X|S=1}[\hat{\eta}(X,1)]}{\mathbb{E}_{X|S=1}[\hat{\eta}(X,1)]} - 1\right|$$

Clearly $\left|\mathbb{P}(S=1) - \hat{\mathbb{P}}(S=1)\right|$ goes to zero in expectation thanks to the law of large numbers or its finite sample variants. Besides, the term $\left|\frac{\hat{\mathbb{E}}_{X|S=0}[\hat{\eta}(X,0)]}{\mathbb{E}_{X|S=0}[\eta(X,0)]} - 1\right|$ can be seen in the following manner: let $Z \in [0,1]$ be a random variable with law $\mathbb{P}_Z$ and $Z_1, \ldots, Z_M$ be its i.i.d. realization, then sequentially our question is about

$$\left|1 - \frac{\bar{Z}}{\mathbb{E}[Z]}\right| \quad ,$$

with $\bar{Z} = \frac{1}{M}\sum_{i=1}^M Z_i$. This term converges to zero in expectation thanks to the multiplicative Chernoff inequality, which is an exponential concentration inequality that allows to obtain even a rate. Actually, even without the multiplicative Chernoff bound this term goes to zero thanks to the law of large numbers. Therefore, for convergence it remains to study the term

$$(\star) = \left|(\mathbb{E}_{X|S=1} - \hat{\mathbb{E}}_{X|S=1})\left(\mathbb{P}(S=1)(2\hat{\eta}(X,1)-1) - \frac{\hat{\theta}\hat{\eta}(X,1)}{\mathbb{E}_{X|S=1}[\hat{\eta}(X,1)]}\right)_+\right| \quad .$$

Notice that thanks to the second part of Assumption 4.1 and the fact that $\hat{\theta} \in [-2,2]$ we have

$$\left|\frac{\hat{\theta}\hat{\eta}(X,1)}{\mathbb{E}_{X|S=1}[\hat{\eta}(X,1)]}\right| \leq \frac{2}{c_{n,N}} \quad .$$

Therefore, we can upper bound $(\star)$ as

$$(\star) \leq \sup_{t \in [-2/c_{n,N}, 2/c_{n,N}]} \left|(\mathbb{E}_{X|S=1} - \hat{\mathbb{E}}_{X|S=1})(\mathbb{P}(S=1)(2\hat{\eta}(X,1)-1) + t)_+\right| \quad ,$$

where the random quantity has been "supped-out". Introduce,

$$\mathcal{D}_{N_1} = \{X_i \in \mathcal{D}_N : S_i = 1\}$$
$$\mathcal{D}_{N_0} = \{X_i \in \mathcal{D}_N : S_i = 0\} \quad ,$$

of size $N_1$ and $N_0$ respectively, such that $N_1 + N_0 = N$. Clearly we have $\mathcal{D}_{N_s} \overset{\text{i.i.d.}}{\sim} \mathbb{P}_{X|S=s}$ for each $s \in \{0,1\}$. Also recall that Remark B.1 implies that neither $N_0$ nor $N_1$ are equal to zero, however,

both are still random. Besides, denote by $\mathcal{D}_N^S = \{S_i \ : \ (X_i, S_i) \in \mathcal{D}_N\}$ the dataset which is obtained from $\mathcal{D}_N$ by removing features. Thus,

$$\mathbb{E}_{(\mathcal{D}_N)}(\star) \leq \mathbb{E}_{\mathcal{D}_N^S} \mathbb{E}_{\mathcal{D}_{N_1}} \sup_{t \in [-2/c_{n,N}, 2/c_{n,N}]} \left| (\mathbb{E}_{X|S=1} - \hat{\mathbb{E}}_{X|S=1}) \left( (2\hat{\eta}(X, 1) - 1)\mathbb{P}(S = 1) + t \right)_+ \right| \ .$$

Conditionally on $\mathcal{D}_N^S$ we can view $N_0$ and $N_1$ as fixed strictly positive integers, moreover, conditionally on $\mathcal{D}_n$ the estimator $\hat{\eta}$ is not random as it is built *only* on $\mathcal{D}_n$. Thus, we would like to control the following process

$$\mathbb{E}_{\mathcal{D}_{N_1}} \sup_{t \in [-2/c_{n,N}, 2/c_{n,N}]} \left| (\mathbb{E}_{X|S=1} - \hat{\mathbb{E}}_{X|S=1}) \left( (2\hat{\eta}(X, 1) - 1)\mathbb{P}(S = 1) + t \right)_+ \right| \ ,$$

conditionally on $\mathcal{D}_N^S, \mathcal{D}_n$. First of all we rewrite this process as

$$\frac{1}{c_{n,N}} \mathbb{E}_{\mathcal{D}_{N_1}} \sup_{|t| \leq 1} \left| (\mathbb{E}_{X|S=1} - \hat{\mathbb{E}}_{X|S=1}) \left( (2\hat{\eta}(X, 1) - 1)\mathbb{P}(S = 1)c_{n,N} + 2t \right)_+ \right| \ .$$

Thanks to the symmetrization argument we can write

$$\mathbb{E}_{\mathcal{D}_{N_1}} \sup_{|t| \leq 1} \left| (\mathbb{E}_{X|S=1} - \hat{\mathbb{E}}_{X|S=1}) \left( (2\hat{\eta}(X, 1) - 1)\mathbb{P}(S = 1)c_{n,N} + 2t \right)_+ \right|$$

$$\leq 2\mathbb{E}_{\mathcal{D}_{N_1}} \sup_{|t| \leq 1} \left| \frac{1}{N_1} \sum_{i=1}^{N_1} \varepsilon_i f_t(X_i) \right| \ ,$$

where $f_t(\cdot) = \left( (2\hat{\eta}(\cdot, 1) - 1)\mathbb{P}(S = 1)c_{n,N} + 2t \right)_+$. Notice that for each $t, t'$ we have for all $x \in \mathbb{R}^d$

$$|f_t(x) - f_{t'}(x)| \leq 2 |t - t'| \ ,$$

that is, the parametrization is 2-Lipschitz. Therefore, standard results in empirical processes (combine [44, Lemma 6.2] with [29, Theorem 3.2.]) tells us that there exists $C > 0$ such that

$$\mathbb{E}_{\mathcal{D}_{N_1}} \sup_{|t| \leq 1} \left| \frac{1}{N_1} \sum_{i=1}^{N_1} \varepsilon_i f_t(X_i) \right| \leq C\sqrt{\frac{1}{N_1}} \ .$$

Thus, taking expectation *w.r.t.* $\mathcal{D}_N^s$ we get

$$\mathbb{E}_{(\mathcal{D}_N)}(\star) \leq \frac{C}{c_{n,N}} \mathbb{E}_{\mathcal{D}_N^s} \sqrt{\frac{1}{N_1}} \ ,$$

applying Lemma B.2 we get for some $C > 0$ that depends on $\mathbb{P}(S = 1)$ that

$$\mathbb{E}_{(\mathcal{D}_N)}(\star) \leq \frac{C}{c_{n,N}} \sqrt{\frac{1}{N}} \ .$$

Thanks to Assumption 4.1 we have

$$\frac{1}{c_{n,N}\sqrt{N}} = o\,(1) \ ,$$

thus, the term $\mathbb{E}_{(\mathcal{D}_N)}(\star)$ converges to zero. Repeating the same argument for $(\triangle\triangle)$ we conclude. $\quad\square$

# D   Proofs of auxiliary results

*Proof of Lemma C.1.* We start from the level of unfairness of $g$, that is, we would like to find an upper bound on

$$|\mathbb{P}\,(g(X, S) = 1 \,|\, S = 1, Y = 1) - \mathbb{P}\,(g(X, S) = 1 \,|\, S = 0, Y = 1)| \ ,$$

rewriting the expression above, our goal can be written as

$$\left| \frac{\mathbb{E}_{X|S=1}\eta(X, 1)g(X, 1)}{\mathbb{E}_{X|S=1}\eta(X, 1)} - \frac{\mathbb{E}_{X|S=0}\eta(X, 0)g(X, 0)}{\mathbb{E}_{X|S=0}\eta(X, 0)} \right| \ .$$

Now, we start working with the expression above

$$\left| \frac{\mathbb{E}_{X|S=1}\eta(X,1)g(X,1)}{\mathbb{E}_{X|S=1}\eta(X,1)} - \frac{\mathbb{E}_{X|S=0}\eta(X,0)g(X,0)}{\mathbb{E}_{X|S=0}\eta(X,0)} \right|$$

$$\leq \left| \frac{\mathbb{E}_{X|S=1}\eta(X,1)g(X,1)}{\mathbb{E}_{X|S=1}\eta(X,1)} - \frac{\mathbb{E}_{X|S=1}\hat{\eta}(X,1)g(X,1)}{\mathbb{E}_{X|S=1}\hat{\eta}(X,1)} \right|$$

$$+ \left| \frac{\mathbb{E}_{X|S=0}\hat{\eta}(X,0)g(X,0)}{\mathbb{E}_{X|S=0}\hat{\eta}(X,0)} - \frac{\mathbb{E}_{X|S=0}\eta(X,0)g(X,0)}{\mathbb{E}_{X|S=0}\eta(X,0)} \right|$$

$$+ \left| \frac{\mathbb{E}_{X|S=1}\hat{\eta}(X,1)g(X,1)}{\mathbb{E}_{X|S=1}\hat{\eta}(X,1)} - \frac{\mathbb{E}_{X|S=0}\hat{\eta}(X,0)g(X,0)}{\mathbb{E}_{X|S=0}\hat{\eta}(X,0)} \right| \;.$$

The first two terms on the right hand side of the inequality can be upper-bounded in a similar way. That is why we only show the bound for the first term, that is, for $S = 1$. We have for $(*) = \left| \frac{\mathbb{E}_{X|S=1}\eta(X,1)g(X,1)}{\mathbb{E}_{X|S=1}\eta(X,1)} - \frac{\mathbb{E}_{X|S=1}\hat{\eta}(X,1)g(X,1)}{\mathbb{E}_{X|S=1}\hat{\eta}(X,1)} \right|$

$$(*) \leq \frac{\mathbb{E}_{X|S=1}|\eta(X,1)-\hat{\eta}(X,1)|}{\mathbb{P}(Y=1\,|\,S=1)} + \left| \frac{\mathbb{E}_{X|S=1}\hat{\eta}(X,1)g(X,1)}{\mathbb{E}_{X|S=1}\eta(X,1)} - \frac{\mathbb{E}_{X|S=1}\hat{\eta}(X,1)g(X,1)}{\mathbb{E}_{X|S=1}\hat{\eta}(X,1)} \right|$$

$$\leq \frac{\mathbb{E}_{X|S=1}|\eta(X,1)-\hat{\eta}(X,1)|}{\mathbb{P}(Y=1\,|\,S=1)}$$

$$+ \mathbb{E}_{X|S=1}\hat{\eta}(X,1)g(X,1) \left| \frac{\mathbb{E}_{X|S=1}\hat{\eta}(X,1)}{\mathbb{E}_{X|S=1}\eta(X,1)\mathbb{E}_{X|S=1}\hat{\eta}(X,1)} - \frac{\mathbb{E}_{X|S=1}\eta(X,1)}{\mathbb{E}_{X|S=1}\hat{\eta}(X,1)\mathbb{E}_{X|S=1}\eta(X,1)} \right|$$

$$\leq \frac{\mathbb{E}_{X|S=1}|\eta(X,1)-\hat{\eta}(X,1)|}{\mathbb{P}(Y=1\,|\,S=1)} + \mathbb{E}_{X|S=1}\hat{\eta}(X,1)g(X,1)\frac{\mathbb{E}_{X|S=1}|\hat{\eta}(X,1)-\hat{\eta}(X,1)|}{\mathbb{E}_{X|S=1}\eta(X,1)\mathbb{E}_{X|S=1}\hat{\eta}(X,1)}$$

$$\leq 2\frac{\mathbb{E}_{X|S=1}|\eta(X,1)-\hat{\eta}(X,1)|}{\mathbb{P}(Y=1\,|\,S=1)} \;,$$

thus, we have

$$|\mathbb{P}(g(X,S)=1\,|\,S=1,Y=1) - \mathbb{P}(g(X,S)=1\,|\,S=0,Y=1)|$$

$$\leq 2\frac{\mathbb{E}_{X|S=1}|\eta(X,1)-\hat{\eta}(X,1)|}{\mathbb{P}(Y=1\,|\,S=1)}$$

$$+ 2\frac{\mathbb{E}_{X|S=0}|\eta(X,0)-\hat{\eta}(X,0)|}{\mathbb{P}(Y=1\,|\,S=0)}$$

$$+ \left| \frac{\mathbb{E}_{X|S=1}\hat{\eta}(X,1)g(X,1)}{\mathbb{E}_{X|S=1}\hat{\eta}(X,1)} - \frac{\mathbb{E}_{X|S=0}\hat{\eta}(X,0)g(X,0)}{\mathbb{E}_{X|S=0}\hat{\eta}(X,0)} \right| \;.$$

Finally, it remains to upper bound

$$(**) = \left| \frac{\mathbb{E}_{X|S=1}\hat{\eta}(X,1)g(X,1)}{\mathbb{E}_{X|S=1}\hat{\eta}(X,1)} - \frac{\mathbb{E}_{X|S=0}\hat{\eta}(X,0)g(X,0)}{\mathbb{E}_{X|S=0}\hat{\eta}(X,0)} \right| \;.$$

Recall that $\hat{\mathbb{E}}_{X|S=1}$ and $\hat{\mathbb{E}}_{X|S=0}$ stands for the expectations taken *w.r.t.* empirical measure induced by $\mathcal{D}_N$, and that $\mathcal{D}_N$ is independent from $\mathcal{D}_n$. Therefore, we can write

$$(**) \leq \left| \frac{\mathbb{E}_{X|S=1}\hat{\eta}(X,1)g(X,1)}{\mathbb{E}_{X|S=1}\hat{\eta}(X,1)} - \frac{\hat{\mathbb{E}}_{X|S=1}\hat{\eta}(X,1)g(X,1)}{\hat{\mathbb{E}}_{X|S=1}\hat{\eta}(X,1)} \right|$$

$$+ \left| \frac{\mathbb{E}_{X|S=0}\hat{\eta}(X,0)g(X,0)}{\mathbb{E}_{X|S=0}\hat{\eta}(X,0)} - \frac{\hat{\mathbb{E}}_{X|S=0}\hat{\eta}(X,0)g(X,0)}{\hat{\mathbb{E}}_{X|S=0}\hat{\eta}(X,0)} \right|$$

$$+ \left| \frac{\hat{\mathbb{E}}_{X|S=1}\hat{\eta}(X,1)g(X,1)}{\hat{\mathbb{E}}_{X|S=1}\hat{\eta}(X,1)} - \frac{\hat{\mathbb{E}}_{X|S=0}\hat{\eta}(X,0)g(X,0)}{\hat{\mathbb{E}}_{X|S=0}\hat{\eta}(X,0)} \right| \;.$$

Clearly, the last term on the right hand side of the previous inequality corresponds to our empirical criteria since everything can be easily evaluated using data. The first two terms on the right hand side

of the inequality can be upper-bounded in a similar fashion, again, we only demonstrate the bound for $S = 1$. We can write

$$\left| \frac{\mathbb{E}_{X|S=1}\hat{\eta}(X,1)g(X,1)}{\mathbb{E}_{X|S=1}\hat{\eta}(X,1)} - \frac{\hat{\mathbb{E}}_{X|S=1}\hat{\eta}(X,1)g(X,1)}{\hat{\mathbb{E}}_{X|S=1}\hat{\eta}(X,1)} \right|$$

$$\leq \left| \frac{\mathbb{E}_{X|S=1}\hat{\eta}(X,1)g(X,1)}{\mathbb{E}_{X|S=1}\hat{\eta}(X,1)} - \frac{\hat{\mathbb{E}}_{X|S=1}\hat{\eta}(X,1)g(X,1)}{\mathbb{E}_{X|S=1}\hat{\eta}(X,1)} \right|$$

$$+ \left| \frac{\hat{\mathbb{E}}_{X|S=1}\hat{\eta}(X,1)g(X,1)}{\mathbb{E}_{X|S=1}\hat{\eta}(X,1)} - \frac{\hat{\mathbb{E}}_{X|S=1}\hat{\eta}(X,1)g(X,1)}{\hat{\mathbb{E}}_{X|S=1}\hat{\eta}(X,1)} \right| \quad .$$

Notice that for the first term on the right hand side of the inequality we have

$$\left| \frac{\mathbb{E}_{X|S=1}\hat{\eta}(X,1)g(X,1)}{\mathbb{E}_{X|S=1}\hat{\eta}(X,1)} - \frac{\hat{\mathbb{E}}_{X|S=1}\hat{\eta}(X,1)g(X,1)}{\mathbb{E}_{X|S=1}\hat{\eta}(X,1)} \right| \leq \frac{\left| \mathbb{E}_{X|S=1}\hat{\eta}(X,1)g(X,1) - \hat{\mathbb{E}}_{X|S=1}\hat{\eta}(X,1)g(X,1) \right|}{\mathbb{E}_{X|S=1}\hat{\eta}(X,1)} \quad ,$$

whereas for the second term we can write

$$\left| \frac{\hat{\mathbb{E}}_{X|S=1}\hat{\eta}(X,1)g(X,1)}{\mathbb{E}_{X|S=1}\hat{\eta}(X,1)} - \frac{\hat{\mathbb{E}}_{X|S=1}\hat{\eta}(X,1)g(X,1)}{\hat{\mathbb{E}}_{X|S=1}\hat{\eta}(X,1)} \right| \leq \frac{\left| \hat{\mathbb{E}}_{X|S=1}\hat{\eta}(X,1) - \mathbb{E}_{X|S=1}\hat{\eta}(X,1) \right|}{\mathbb{E}_{X|S=1}\hat{\eta}(X,1)} \quad .$$

$\square$

*Proof of Lemma C.2.* Let us first introduce two slices of $\mathcal{D}_N$ as

$$\mathcal{D}_{N_1} = \{ X_i \in \mathcal{D}_N \; : \; S_i = 1 \}, \; \mathcal{D}_{N_0} = \{ X_i \in \mathcal{D}_N \; : \; S_i = 0 \}$$

of size $N_1$ and $N_0$ respectively, such that $N_1 + N_0 = N$. Clearly we have $\mathcal{D}_{N_s} \overset{\text{i.i.d.}}{\sim} \mathbb{P}_{X|S=s}$ for each $s \in \{0,1\}$. Besides, denote by $\mathcal{D}_N^S = \{ S_i \; : \; (X_i, S_i) \in \mathcal{D}_N \}$ the which is obtained from $\mathcal{D}_N$ by removing features. Recalling Remark B.1, we have

$$N_1 - 2 \sim \text{Bin}(N, \mathbb{P}(S = 1)), \quad N_0 - 2 \sim \text{Bin}(N, \mathbb{P}(S = 0)) \quad .$$

Clearly, since the proposed algorithm is a thresholding of $\hat{\eta}$ we have

$$\mathbb{E}_{(\mathcal{D}_n, \mathcal{D}_N)} \left| (\mathbb{E}_{X|S=0} - \hat{\mathbb{E}}_{X|S=0})\hat{\eta}(X,0)\hat{g}(X,0) \right|$$

$$\leq \mathbb{E}_{(\mathcal{D}_n, \mathcal{D}_N)} \sup_{t \in [0,1]} \left| (\mathbb{E}_{X|S=0} - \hat{\mathbb{E}}_{X|S=0})\hat{\eta}(X,0)\mathbf{1}_{\{t \leq \hat{\eta}(X,0)\}} \right| \quad .$$

Further we work conditionally on $\mathcal{D}_n$. Using the classical symmetrization technique [29, Theorem 2.1.] we get

$$\mathbb{E}_{\mathcal{D}_N} \sup_{t \in [0,1]} \left| (\mathbb{E}_{X|S=0} - \hat{\mathbb{E}}_{X|S=0})\hat{\eta}(X,0)\mathbf{1}_{\{t \leq \hat{\eta}(X,0)\}} \right|$$

$$= \mathbb{E}_{\mathcal{D}_N^S} \mathbb{E}_{\mathcal{D}_{N_0}} \sup_{t \in [0,1]} \left| (\mathbb{E}_{X|S=0} - \hat{\mathbb{E}}_{X|S=0})\hat{\eta}(X,0)\mathbf{1}_{\{t \leq \hat{\eta}(X,0)\}} \right|$$

$$\leq 2\mathbb{E}_{\mathcal{D}_N^S} \mathbb{E}_{\mathcal{D}_{N_0}} \mathbb{E}_\varepsilon \sup_{t \in [0,1]} \left| \frac{1}{N_0} \sum_{X_i \in \mathcal{D}_{N_0}} \varepsilon_i \hat{\eta}(X_i,0)\mathbf{1}_{\{t \leq \hat{\eta}(X_i,0)\}} \right| \quad ,$$

where $\varepsilon_i \overset{\text{i.i.d.}}{\sim}$ Rademacher variables. Note that the function class $x \mapsto \mathbf{1}_{\{t \leq \hat{\eta}(x,0)\}}$ has VC-dimension [43] equal to one. At this moment we will work with

$$\mathbb{E}_\varepsilon \sup_{t \in [0,1]} \left| \frac{1}{N_0} \sum_{X_i \in \mathcal{D}_{N_0}} \varepsilon_i \hat{\eta}(X_i,0)\mathbf{1}_{\{t \leq \hat{\eta}(X_i,0)\}} \right| \quad ,$$

conditionally on all the data. First of all let us introduce $\mathcal{F} = \left\{ f \ : \ \exists t \in [0,1], \ f(x) = \mathbf{1}_{\{t \leq \hat{\eta}(x,0)\}} \right\}$
Thus, our process can be written as

$$\mathbb{E}_{\varepsilon} \sup_{f \in \mathcal{F}} \left| \frac{1}{N_0} \sum_{X_i \in \mathcal{D}_{N_0}} \varepsilon_i \varphi_i(f(X_i)) \right| \ ,$$

where $\varphi_i(\cdot) = \eta(X_i, 0) \times \cdot$. Clearly, we have $\varphi_i(0) = 0$ and for every $u, v$

$$|\varphi_i(u) - \varphi_i(v)| \leq |u - v| \ .$$

That is, $\varphi_i$ are contractions, and the contraction lemma [29, Theorem 2.2.] gives

$$\mathbb{E}_{\varepsilon} \sup_{f \in \mathcal{F}} \left| \frac{1}{N_0} \sum_{X_i \in \mathcal{D}_{N_0}} \varepsilon_i \varphi_i(f(X_i)) \right| \leq \mathbb{E}_{\varepsilon} \sup_{f \in \mathcal{F}} \left| \frac{1}{N_0} \sum_{X_i \in \mathcal{D}_{N_0}} \varepsilon_i f(X_i) \right| \ .$$

Recall, that the class $\mathcal{F}$ is a VC-class with VC-dimension equal to one. Therefore, it is a known fact [18, 34] that there exists $C > 0$ such that

$$\mathbb{E}_{\varepsilon} \sup_{f \in \mathcal{F}} \left| \frac{1}{N_0} \sum_{X_i \in \mathcal{D}_{N_0}} \varepsilon_i f(X_i) \right| \leq C \sqrt{\frac{1}{N_0}} \ ,$$

almost surely. The above implies that

$$\mathbb{E}_{\mathcal{D}_N} \sup_{t \in [0,1]} \left| (\mathbb{E}_{X|S=0} - \hat{\mathbb{E}}_{X|S=0}) \hat{\eta}(X, 0) \mathbf{1}_{\{t \leq \hat{\eta}(X,0)\}} \right| \leq C \mathbb{E}_{\mathcal{D}_N^S} \sqrt{\frac{1}{N_0}} \ .$$

It remains to provide an upper bound on $\mathbb{E}_{\mathcal{D}_N^S} \sqrt{\frac{1}{N_0}}$, to this end we recall that this expectation can be written as

$$\mathbb{E} \sqrt{\frac{1}{2 + Z}} \ ,$$

where $Z$ is the binomial random variable with parameters $N$ and $\mathbb{P}(S = 0)$. Thus, thanks to Lemma B.2 there exists a constant $C > 0$ that depends on $\mathbb{P}(S = 0)$ such that

$$\mathbb{E} \sqrt{\frac{1}{2 + Z}} \leq C \sqrt{\frac{1}{N}} \ .$$

Similarly we get the bound for the case $S = 1$. $\qquad\square$

# E   Optimal classifier independent of sensitive feature

In this section we provide guidelines to construct a *plug-in* algorithm which can use the sensitive feature only at training time but cannot use it for future decision making. It is clear that the first step would be to derive fair optimal classifier $g^* : \mathbb{R}^d \to \{0, 1\}$ which is defined as

$$g^* \in \arg\min \left\{ \mathcal{R}(g) \ : \ \mathbb{P}\left( g(X) = 1 \,|\, S = 1, Y = 1 \right) = \mathbb{P}\left( g(X) = 1 \,|\, S = 0, Y = 1 \right) \right\} \ ,$$

with $\mathcal{R}(g) := \mathbb{P}(Y \neq g(X))$. Next result establishes this expression.

**Proposition E.1** (Optimal rule). *Under Assumption 2.2 an optimal classifier $g^*$ can be obtained for all $x \in \mathbb{R}^d$ as*

$$g^*(x) = \mathbf{1}_{\left\{ 1 \leq 2\eta(x) + \theta^* \left( \frac{\eta(x,0)}{\mathbb{E}_X[\eta(X,0)]} - \frac{\eta(x,1)}{\mathbb{E}_X[\eta(X,1)]} \right) \right\}} \ ,$$

*where $\theta^*$ is such that the equality*

$$\frac{\mathbb{E}_X\left[ \eta(X,1) g^*(X) \right]}{\mathbb{E}_X[\eta(X,1)]} = \frac{\mathbb{E}_X\left[ \eta(X,0) g^*(X) \right]}{\mathbb{E}_X[\eta(X,0)]} \ ,$$

*is satisfied and $\eta(\cdot) := \mathbb{P}\left( Y = 1 \,|\, X = \cdot \right)$.*

Observe that to efficiently compute the optimal classifier in this case we need to have access to $\eta(x), \eta(x, s)$ and marginal distribution $\mathbb{P}_X$.

This observation motivates us to propose a plug-in algorithm based on two datasets $\mathcal{D}_n = \{(X_i, S_i, Y_i)\}_{i=1}^n$ and $\mathcal{D}_N = \{X_i\}_{i=1}^N$. The labeled data $\mathcal{D}_n$ allow to estimate $\eta(x), \eta(x, s)$ and the unlabeled data $\mathcal{D}_N$ allow to estimate the marginal distribution $\mathbb{P}_X$. Interestingly, we do not need to observe sensitive features in the unlabeled dataset $\mathcal{D}_N$.

Formally, our procedure $\hat{g}$ in this case can be defined for all $x \in \mathbb{R}^d$ as

$$\hat{g}(x) = \mathbf{1}_{\left\{1 \leq 2\hat{\eta}(x) + \hat{\theta}\left(\frac{\hat{\eta}(x,0)}{\hat{\mathbb{E}}_X[\hat{\eta}(X,0)]} - \frac{\hat{\eta}(x,1)}{\hat{\mathbb{E}}_X[\eta(X,1)]}\right)\right\}} \, ,$$

where $\hat{\eta}(x), \hat{\eta}(x, s)$ for all $s \in \{0, 1\}$ are the estimates of regression functions constructed on $\mathcal{D}_n$, and $\hat{\mathbb{E}}_X$ is the empirical expectation based on $\mathcal{D}_N$.

Finally, similarly to the previous case the threshold $\hat{\theta}$ is defined as

$$\hat{\theta} \in \arg\min_{\theta} \left| \frac{\hat{\mathbb{E}}_X[\hat{\eta}(X,1)\hat{g}_\theta(X)]}{\hat{\mathbb{E}}_X[\hat{\eta}(X,1)]} - \frac{\hat{\mathbb{E}}_X[\hat{\eta}(X,0)\hat{g}_\theta(X)]}{\hat{\mathbb{E}}_X[\hat{\eta}(X,0)]} \right| \, ,$$

with $\hat{g}_\theta$ defined for all $x \in \mathbb{R}^d$ as

$$\hat{g}_\theta(x) = \mathbf{1}_{\left\{1 \leq 2\hat{\eta}(x) + \theta\left(\frac{\hat{\eta}(x,0)}{\hat{\mathbb{E}}_X[\hat{\eta}(X,0)]} - \frac{\hat{\eta}(x,1)}{\hat{\mathbb{E}}_X[\eta(X,1)]}\right)\right\}} \, .$$

### E.1 Proofs

*Proof of Proposition E.1.* Let us study the following minimization problem

$$(*) := \min_{g \in \mathcal{G}} \{\mathcal{R}(g) \, : \, \mathbb{P}\left(g(X) = 1 \,|\, Y = 1, S = 1\right) = \mathbb{P}\left(g(X) = 1 \,|\, Y = 1, S = 0\right)\} \, .$$

Using the weak duality we can write

$$(*) = \min_{g \in \mathcal{G}} \max_{\lambda \in \mathbb{R}} \{\mathcal{R}(g) + \lambda\left(\mathbb{P}\left(g(X) = 1 \,|\, Y = 1, S = 1\right) - \mathbb{P}\left(g(X) = 1 \,|\, Y = 1, S = 0\right)\right)\}$$

$$\geq \max_{\lambda \in \mathbb{R}} \min_{g \in \mathcal{G}} \{\mathcal{R}(g) + \lambda\left(\mathbb{P}\left(g(X) = 1 \,|\, Y = 1, S = 1\right) - \mathbb{P}\left(g(X) = 1 \,|\, Y = 1, S = 0\right)\right)\}$$

$$=: (**) \, .$$

We first study the objective function of the max min problem $(**)$, which is equal to

$$\mathbb{P}(g(X) \neq Y) + \lambda\left(\mathbb{P}\left(g(X) = 1 \,|\, Y = 1, S = 1\right) - \mathbb{P}\left(g(X) = 1 \,|\, Y = 1, S = 0\right)\right) \, .$$

Using arguments of Lemma B.3 we can write

$$\mathbb{P}(g(X) \neq Y) = \mathbb{P}(Y = 1) - \mathbb{E}_X[(2\eta(X) - 1)g(X)] \, ,$$

where $\eta(\cdot) := \mathbb{P}(Y = 1 | X = \cdot)$. Moreover, since

$$\mathbb{E}[YS] = \mathbb{E}_S[S\mathbb{E}[Y|S]] = \mathbb{E}_S[S\mathbb{E}_X[\mathbb{E}[Y|X,S]]] = \mathbb{E}_S[S\mathbb{E}_X[\eta(X,S)]] = \mathbb{P}(S = 1)\mathbb{E}_X[\eta(X,1)] \, ,$$

we can write for the rest

$$\mathbb{P}\left(g(X) = 1 \,|\, Y = 1, S = 1\right) = \frac{\mathbb{P}\left(g(X) = 1, Y = 1, S = 1\right)}{\mathbb{P}\left(Y = 1, S = 1\right)} = \frac{\mathbb{E}[g(X)YS]}{\mathbb{E}[YS]}$$

$$= \frac{\mathbb{P}(S = 1)\mathbb{E}_X[g(X)\eta(X,1)]}{\mathbb{P}(S = 1)\mathbb{E}_X[\eta(X,1)]} = \frac{\mathbb{E}_X[g(X)\eta(X,1)]}{\mathbb{E}_X[\eta(X,1)]}$$

$$\mathbb{P}\left(g(X) = 1 \,|\, Y = 1, S = 0\right) = \frac{\mathbb{P}\left(g(X) = 1, Y = 1, S = 0\right)}{\mathbb{P}\left(Y = 1, S = 0\right)} = \frac{\mathbb{E}[g(X)Y(1 - S)]}{\mathbb{E}[Y(1 - S)]}$$

$$= \frac{\mathbb{E}_X[g(X)\eta(X,0)]}{\mathbb{E}_X[\eta(X,0)]} \, .$$

Using these, the objective of $(**)$ can be simplified as

$$\mathbb{P}(Y = 1) - \mathbb{E}_X\left[g(X)\left(2\eta(X) - 1 + \lambda\left(\frac{\eta(X,0)}{\mathbb{E}_X[\eta(X,0)]} - \frac{\eta(X,1)}{\mathbb{E}_X[\eta(X,1)]}\right)\right)\right] \, .$$

Clearly, for every $\lambda \in \mathbb{R}$ a minimizer $g_\lambda^*$ of the problem $(**)$ can be written for all $x \in \mathbb{R}^d$ as

$$g_\lambda^*(x) = \mathbf{1}_{\left\{ 2\eta(x)-1+\lambda\left( \frac{\eta(x,0)}{\mathbb{E}_X[\eta(X,0)]} - \frac{\eta(x,1)}{\mathbb{E}_X[\eta(X,1)]} \right) \geq 0 \right\}} \ .$$

Similarly to Proposition 2.3, for $\lambda = 0$ we recover the classical optimal predictor in the context of binary classification. Substituting this classifier into the objective of $(**)$ we arrive at

$$(**) = \mathbb{P}(Y=1) - \min_{\lambda \in \mathbb{R}} \left\{ \mathbb{E}_X \left( 2\eta(X) - 1 + \lambda \left( \frac{\eta(X,0)}{\mathbb{E}_X[\eta(X,0)]} - \frac{\eta(X,1)}{\mathbb{E}_X[\eta(X,1)]} \right) \right)_+ \right\} \ .$$

The mapping

$$\lambda \mapsto \mathbb{E}_X \left( 2\eta(X) - 1 + \lambda \left( \frac{\eta(X,0)}{\mathbb{E}_X[\eta(X,0)]} - \frac{\eta(X,1)}{\mathbb{E}_X[\eta(X,1)]} \right) \right)_+ \ ,$$

is convex, therefore we can write the first order optimality conditions as

$$0 \in \partial_\lambda \mathbb{E}_X \left( 2\eta(X) - 1 + \lambda \left( \frac{\eta(X,0)}{\mathbb{E}_X[\eta(X,0)]} - \frac{\eta(X,1)}{\mathbb{E}_X[\eta(X,1)]} \right) \right)_+ \ .$$

Clearly, under continuity assumption this subgradient is reduced to the gradient almost surely, thus we have the following condition on the optimal value of $\lambda^*$

$$\frac{\mathbb{E}_X\left[\eta(X,1)g_{\lambda^*}^*(X)\right]}{\mathbb{E}_X[\eta(X,1)]} = \frac{\mathbb{E}_X\left[\eta(X,0)g_{\lambda^*}^*(X)\right]}{\mathbb{E}_X[\eta(X,0)]} \ ,$$

and the pair $(\lambda^*, g_{\lambda^*}^*)$ is a solution of the dual problem $(**)$. Notice that the previous condition can be written as

$$\mathbb{P}\left(g_{\lambda^*}^*(X) = 1 \mid Y=1, S=1\right) = \mathbb{P}\left(g_{\lambda^*}^*(X) = 1 \mid Y=1, S=0\right) \ .$$

This implies that the classifier $g_{\lambda^*}^*$ is fair. Finally, it remains to show that $g_{\lambda^*}^*$ is actually an optimal classifier, indeed, since $g_{\lambda^*}^*$ is fair we can write on the one hand

$$\mathcal{R}(g_{\lambda^*}^*) \geq \min_{g \in \mathcal{G}} \left\{ \mathcal{R}(g) \ : \ \mathbb{P}\left(g(X)=1 \mid Y=1, S=1\right) = \mathbb{P}\left(g(X)=1 \mid Y=1, S=0\right) \right\} = (*).$$

On the other hand the pair $(\lambda^*, g_{\lambda^*}^*)$ is a solution of the dual problem $(**)$, thus we have

$$\begin{aligned}
(*) &\geq \mathcal{R}(g_{\lambda^*}^*) + \lambda^* \left( \mathbb{P}\left(g_{\lambda^*}^*(X)=1 \mid Y=1, S=1\right) - \mathbb{P}\left(g_{\lambda^*}^*(X)=1 \mid Y=1, S=0\right) \right) \\
&= \mathcal{R}(g_{\lambda^*}^*) \ .
\end{aligned}$$

It implies that the classifier $g_{\lambda^*}^*$ is optimal, hence $g^* \equiv g_{\lambda^*}^*$. $\qquad\square$

## E.2 Experiments without the sensitive feature

In this section we report the equivalent results to those in Table 1 and Figure 1 into Table 3 and Figure 2 when the sensitive feature is not in the functional form of the model. Note that the method of Hardt [22] is not able to deal with this setting then there are no results for this case.

From Table 3 and Figure 2 we can observe analogous results to those in Section 5. Nevertheless, note that, without the sensitive feature in the functional form of the models, the results are generally less accurate and more fair w.r.t. to the case that the sensitive feature in the functional form of the models. This results is similar to the one reported in [17].

| Method | Arrhythmia ACC | Arrhythmia DEO | COMPAS ACC | COMPAS DEO | Adult ACC | Adult DEO | German ACC | German DEO | Drug ACC | Drug DEO |
|---|---|---|---|---|---|---|---|---|---|---|
| Lin.SVM | 0.71±0.05 | 0.10±0.03 | 0.72±0.01 | 0.12±0.02 | 0.78 | 0.09 | 0.69±0.04 | 0.11±0.10 | 0.79±0.02 | 0.25±0.04 |
| Lin.LR | 0.71±0.04 | 0.11±0.04 | 0.73±0.02 | 0.10±0.03 | 0.80 | 0.08 | 0.68±0.05 | 0.12±0.09 | 0.80±0.03 | 0.23±0.03 |
| Lin.SVM+Hardt | - | - | - | - | - | - | - | - | - | - |
| Lin.LR+Hardt | - | - | - | - | - | - | - | - | - | - |
| Zafar | 0.67±0.03 | 0.05±0.02 | 0.69±0.01 | 0.10±0.08 | 0.76 | 0.05 | 0.62±0.09 | 0.13±0.10 | 0.66±0.03 | 0.06±0.06 |
| Lin.Donini | 0.75±0.05 | 0.05±0.02 | 0.73±0.01 | 0.07±0.02 | 0.75 | 0.01 | 0.69±0.04 | 0.06±0.03 | 0.79±0.02 | 0.10±0.06 |
| Lin.SVM+Ours | 0.72±0.05 | 0.03±0.01 | 0.72±0.01 | 0.06±0.02 | 0.74 | 0.02 | 0.68±0.04 | 0.06±0.04 | 0.78±0.02 | 0.12±0.02 |
| Lin.LR+Ours | 0.71±0.04 | 0.04±0.02 | 0.71±0.02 | 0.06±0.02 | 0.76 | 0.02 | 0.67±0.05 | 0.05±0.03 | 0.79±0.03 | 0.10±0.01 |
| SVM | 0.71±0.05 | 0.10±0.03 | 0.73±0.01 | 0.11±0.02 | 0.79 | 0.08 | 0.74±0.03 | 0.10±0.06 | 0.81±0.02 | 0.22±0.03 |
| LR | 0.70±0.06 | 0.10±0.03 | 0.74±0.01 | 0.10±0.02 | 0.78 | 0.10 | 0.75±0.03 | 0.09±0.05 | 0.81±0.03 | 0.21±0.02 |
| RF | 0.81±0.02 | 0.08±0.02 | 0.76±0.03 | 0.10±0.02 | 0.84 | 0.11 | 0.77±0.03 | 0.07±0.04 | 0.85±0.02 | 0.19±0.02 |
| SVM+Hardt | - | - | - | - | - | - | - | - | - | - |
| LR+Hardt | - | - | - | - | - | - | - | - | - | - |
| RF+Hardt | - | - | - | - | - | - | - | - | - | - |
| Donini | 0.75±0.05 | 0.05±0.02 | 0.72±0.01 | 0.08±0.02 | 0.77 | 0.01 | 0.73±0.04 | 0.05±0.03 | 0.79±0.03 | 0.10±0.05 |
| SVM+Ours | 0.71±0.02 | 0.06±0.02 | 0.72±0.01 | 0.05±0.02 | 0.78 | 0.02 | 0.73±0.01 | 0.06±0.03 | 0.78±0.02 | 0.11±0.02 |
| LR+Ours | 0.70±0.04 | 0.06±0.03 | 0.72±0.01 | 0.06±0.02 | 0.77 | 0.02 | 0.73±0.02 | 0.06±0.02 | 0.77±0.02 | 0.11±0.02 |
| RF+Ours | 0.80±0.03 | 0.02±0.01 | 0.76±0.02 | 0.04±0.02 | 0.84 | 0.02 | 0.76±0.03 | 0.04±0.02 | 0.83±0.01 | 0.06±0.02 |

Table 3: Results (average $\pm$ standard deviation, when a fixed test set is not provided) for all the datasets, concerning ACC and DEO. In this case the sensitive feature the sensitive feature is not in the functional form of the model.

Figure 2: Results of Table 3 of linear (left) and nonlinear (right) methods when the error and the DEO are normalized in $[0, 1]$ column-wise. Different colors and symbols refer to different datasets and method respectively. The closer a point is to the origin, the better the result is. In this case the sensitive feature the sensitive feature is not in the functional form of the model.