[Reviews · NeurIPS 2019]

Reviewer 1



While I personally am not very excited by yet another algorithm-for-group-fairness-in-binary-classification paper, this work does fill in a real gap in the literature. I did not check proofs carefully, but otherwise this work is pretty well written and appears correct. This work does perform one unfortunate sleight-of-hand by starting with exact fairness and giving Bayes optimal for this exact case, but then only proving that their algorithm is exactly fair in the limit. While exact fairness won't be possible on the distribution, it should be on the sample at least. Relatedly, I would prefer to see the finite sample results they claim to have, but do not show, even if those results are highly dependent on \eta. Assumption 2.2 seems, regardless of whether it's been used before, difficult to buy as a reasonable assumption. In practice, there may only be finitely many values of X with non-zero support, and therefore you'll have non-zero probability mass on the events in question. I would guess that removing the assumption would make the results messier, but it should still be possible to write down without this assumption. Finally, the experiments left me more puzzled than satisfied, as their claim was that their method was preferable to the other top performer because it was more general and could be applied to RF, which was indeed one of the best algorithms on these datasets for minimizing error. But RF is clearly more difficult to get to be fair, because on at least some of the data sets, the RF+post-processing methods were not top performers. Other things: -There are quite a few tyos/grammatical issues, starting with lines 34, 38, 65, 72, 87, 90, 93, etc. -line 88: \mathcal{G} is not a set Edit after author response: Thanks for addressing my concerns in detail.

Reviewer 2



Originality: -Although the Hardt paper has suggested the use of this approach, the paper claims that it’s the first to actually show this is indeed possible. It's the first to decouple the risk and fairness in terms of regression function and base rate. -Use of unlabeled data to help facilitate fairness along with better accuracy is novel. Quality: -The assumptions made about the model are very well justified. The discussion after each assumption provided the context as to why the assumption makes sense and why the assumption is needed to study their model. These discussion as a result provided very good intuition and set up the stage for the proof. -The results in the appendix are quite strong, too. Clarity: -Overall, the paper have a very smooth flow, whether it be discussion of their assumptions or their remarks. -The proof sketch is thorough yet simple enough to understand the gist intuitively. -Once again, the remarks are very helpful. For instance, they expect the natural follow-up questions and answer them very well (e.g. why asymptotic convergence instead of the rate of convergence). Even in the appendix, they provide remarks as to how to read (e.g. make sure to understand some preliminary concepts before proceeding the proof). -The paper follows the neurips guideline very strictly: transparency in terms of hyper parameters are chosen, confidence intervals for the accuracy and fairnes, etc. -It's not a big deal, but the figures are somewhat hard to read; one suggestion is to change the highlight yellow color (Adult Yellow) to something else. Significance: -There's already a huge literature on plug-in approaches. The paper connects the fairness literature with that literature. -The proposed approach of decoupling things into regression function and base rate may be useful for other notions of fairness as well. -Empirically, their methods stay competitive, if not outperform the current state-of-art. -Also, in the appendix, it discusses what the optimal classifier without sensitive feature looks like, although the proof of consistency of the empirical estimator is not provided. Once again, they validate their results in this case empirically. Question: -Isn't it pretty common to assume bounded Rademacher complexity (as in Agarwal et al.) to argue about uniform convergence (generalization error) of the classifier? ***** POST REBUTTAL ****** I am also concerned about the non-trivial changes included in their response, and I'm not too excited about another group fairness paper. However, I think it's still well-written and and techniques are somewhat interesting. Therefore, I'll keep my score at 8.

Reviewer 3



This paper deals with the problem of enforcing Equal Opportunity (Hardt et al.) while leveraging both labeled and unlabeled data. It takes as given a trained black-box classifier and enforces Equal Opportunity on top of that classifier using a threshold rule. Notably, this paper assumes that there are only two sensitive groups, and it is unclear how to extend it to work with multiple groups (see Improvements). Originality: As far as I know, up until now, most methods for enforcing Equal Opportunity (and similar fairness metrics that require labels) have only been able to make use of labeled data at training time. This work provides a novel way to use both labeled and unlabeled data at training time. Quality: The theoretical results in this paper seem good and relevant. It is not ideal that the authors only provide asymptotic results though, and it would be nice to have at least a verbal description about how the amount of unlabeled data N affects the convergence properties. The experiments also appear to be run well with appropriate comparisons to recent methods. However, I would like to see more comparisons with direct constrained optimization approaches that work on nonlinear models, such as Cotter et al. [12] or Agarwal et al [2], since this paper only includes comparison to a direct constrained optimization approach (Zafar [48]) on a linear model. Clarity: The paper is organized clearly with generally easily understandable notation. There are a few 'remarks' which I would prefer to see fleshed out more in the main body of the paper, such as remark 3.3. Significance: In practice, unlabeled data is often easier to come by than labeled data -- in fact, unlabeled data may come easily in very large quantities, possibly orders of magnitude larger than labeled data. Therefore, if this paper is able to effectively leverage unlabeled data to do a better job of enforcing Equal Opportunity, it can have great practical benefits. Unfortunately, the paper only provides asymptotic theoretical results, so it's unclear how much the convergence rate benefits from the amount of unlabeled data (N) compared to the amount of labeled data (n). It would be really nice to see how these two terms interact in the convergence rate (also mentioned in Improvements). Overall, I think this paper provides a novel methodology that adds value over previous methods. The main factors holding me back from a higher score are the lack of theoretical results about the effects of the size of N (e.g. a convergence rate that includes N), and the lack of empirical results that illustrate the effects of varying the size of N, even in the absence of a theoretical convergence rate.

[Author Response · NeurIPS 2019]

We thank all reviewers for their valuable comments. We first address major comments shared by reviewers and then
individual comments.

**Non-asymptotic behavior of the proposed procedure:** In order to present the non-asymptotic result in a clean way
we require few modifications to the paper, which we summarize below.

(1) We slightly modify our proposed procedure in Sect. 3, by incorporating part (ii) in Assumption 4.1 directly into the
method. Specifically, we explicitly perform the truncation proposed in lines 197–200 with $c_{n,N} = N^{-1/4}$.

(2) Assumption 4.1 now consists only of part (i), since part (ii) has been incorporated in the method.

(3) Finally, we remove Remark 4.6 and modify Theorem 4.5, as follows.

**Theorem 4.5.** *Under Assumptions 2.2 and 4.3, there exist universal constants $C, C' > 0$ such that the proposed*
*algorithm (with truncation) satisfies*

$$\mathbb{E}_{(\mathcal{D}_n, \mathcal{D}_N)}[\Delta(\hat{g}, \mathbb{P})] \leq C \sum_{s \in \{0,1\}} \left( \frac{\mathbb{E}_{\mathcal{D}_n}\mathbb{E}_{X|S=s}|\eta(X,s) - \hat{\eta}(X,s)|}{\mathbb{P}(Y=1\,|\,S=s)} + \left(\mathbb{P}(S=s)N\right)^{-1/4} \right) ,$$

$$\mathbb{E}_{(\mathcal{D}_n, \mathcal{D}_N)}[\mathcal{R}(\hat{g})] \leq \mathcal{R}(g^*) + C' \sum_{s \in \{0,1\}} \left( \frac{\mathbb{E}_{\mathcal{D}_n}\mathbb{E}_{X|S=s}|\eta(X,s) - \hat{\eta}(X,s)|}{\mathbb{P}(Y=1\,|\,S=s)} + \left(\mathbb{P}(S=s)N\right)^{-1/4} \right) .$$

*Moreover, if the estimator $\hat{\eta}$ satisfies (modified) Assumption 4.1, the proposed algorithm satisfies*

$$\lim_{n,N\to\infty} \mathbb{E}_{(\mathcal{D}_n, \mathcal{D}_N)}[\Delta(\hat{g}, \mathbb{P})] = 0 \quad and \quad \lim_{n,N\to\infty} \mathbb{E}_{(\mathcal{D}_n, \mathcal{D}_N)}[\mathcal{R}(\hat{g})] \leq \mathcal{R}(g^*) .$$

Let us mention that it is possible to write explicit values for the constants $C, C' > 0$, which are independent from
the parameters of the problem. We can see that the rate *w.r.t* the size of the unlabeled dataset is $N^{-1/4}$. The rate
is non-parametric due to the truncation argument to upper bound the quantity $1/\mathbb{E}_{X|S=s}[\hat{\eta}(X,s)]$. Moreover, let us
mention that even in the presence of an infinite number of unlabeled data, the dependence on the $\ell_1$ norm is unavoidable
for plug-in methods. Indeed, a close inspection of our proof strategy reveals that the pseudo-estimator (see sketch of the
proof of Theorem 4.5) $\tilde{g}$ has this term in its upper bound. Finally, we stress that that in the *classical non-parametric*
classification without extra assumptions, the rate of $\ell_1$ norm estimation of $\eta(\cdot)$ is minimax optimal (see [46]) for the
classification excess risk.

**Extension to several groups:** We note that the argument in Proposition 2.3 extends to the case that the number of
groups $G$ is larger than two. Due to the space limitation, we only sketch the proof using Appendix A as the reference
point. In this general case, the constraints read as

$$\mathbb{P}\left(g(X,S) = 1\,|\,Y=1, S=s\right) = \mathbb{P}\left(g(X,S) = 1\,|\,Y=1, S=s+1\right) \quad \text{for all} \quad s \in \{1, \ldots, G-1\} .$$

For these constraints it is still possible to write the dual problem introducing $\lambda_1, \ldots, \lambda_{G-1}$ real Lagrange multipliers.
Similarly to the proof of Proposition 2.3, we first solve the dual formulation which can be performed explicitly. Unlike
the case of two groups, which results in one condition on one value $\theta^*$, now we will have $G-1$ different conditions for
$G-1$ different values $\theta_1^*, \ldots, \theta_{G-1}^*$. Consequently, once the form of the optimal classifier is established, it will be
apparent how to extend the plug-in approach to this case following our scheme. However, we feel that this extension is
out of the scope of this paper and we prefer to explore it in detail in future work.

**R1.** *"Assumption 2.2 seems, regardless ...".* We agree with the reviewer that theory and practice might do not always
agree with each other. Yet, we care to point out that our theory driven approach shows promising empirical results. In
order to remove this assumption one may consider probabilistic classifiers, which is a valuable future research direction.

*"Finally, the experiments left me more puzzled ...".* Although perhaps we have overstated the good performance of
"RF+Ours", please note that, looking at the results in Table 1, "RF+Ours" is among the best performing methods in
terms of DEO (since DEO should be as small a possible) except for the "Adult" dataset, where the train/test splits were
provided and no cross-validation was performed.

**R2.** *"Isn't it pretty common to assume bounded Rademacher complexity ...".* It is indeed a common assumption
in the study of empirical risk minimizers (ERM). However, there is an important difference between ERM type
algorithms and our plug-in approach. The main goal of ERM theory is to approximate the best classifier in a given
family of classifiers (*e.g.,* linear classifiers), whereas here we directly aim at estimating the optimal (overall) classifier.

**R3.** *"I'm assuming that $\eta$ is the true underlying probability ...".* The reviewer is correct. We agree that the phrasing
might be misleading; we will modify Assumption 2.2 by avoiding the term "regression function".

*"Experiments: given that one of this work ...".* We will address in detail the
comments by the reviewer in the revised version. During the rebuttal we were
able to perform some preliminary experiments on the COMPAS and Adult
dataset, which are the only ones big enough to allow performing the requested
experiments.

| RF+Ours | COMPAS | | Adult | |
|---|---|---|---|---|
| | ACC | DEO | ACC | DEO |
| $\mathcal{D}_n = {}^1/_{10}, \mathcal{D}_N = {}^1/_{10}$ | 0.68 | 0.07 | 0.79 | 0.06 |
| $\mathcal{D}_n = {}^1/_{10}, \mathcal{D}_N = {}^2/_{10}$ | 0.68 | 0.07 | 0.79 | 0.06 |
| $\mathcal{D}_n = {}^1/_{10}, \mathcal{D}_N = {}^4/_{10}$ | 0.70 | 0.06 | 0.79 | 0.05 |
| $\mathcal{D}_n = {}^1/_{10}, \mathcal{D}_N = {}^8/_{10}$ | 0.71 | 0.05 | 0.80 | 0.04 |

*"...I would like to see more comparisons with direct constrained optimization*
*approaches that work on nonlinear models..."* We care to point out that not only
we compared our method with Zafar (which is linear) but also with Donini and Hardt (which works also in the non-linear
case). A comparison to Cotter and Agarwal will be inserted as requested in the revised version but we did not manage
to do it on time for the rebuttal.

**All.** Finally, we thank all the reviewers for their careful reading. We will address all the minor points (typos, notation
issues, figure colors, and remark movements) as underlined and requested by the referees. Of course, we will include
the final not anonymous link to the code upon acceptance.

[Meta-Review · NeurIPS 2019]

Three reviewers who are all good experts for this paper found the paper interesting, novel, compelling, and well-written. With such a difficult topic as fairness, it was particularly helpful that the authors were able to discuss their assumptions, results, and proofs so clearly, and that definitely adds value to the work. The authors' response was appreciated and was found to be helpful, but reviewers expressed some concern in discussion about adding too many new results they didn't have a chance to review, so while we hope the authors can address some of the reviewers suggestions in the final paper, they are encouraged not to add too much stuff that wasn't reviewed, but instead to consider expanding on some of that for a follow-on submission.